# Four ways blue foods can help achieve food system ambitions across nations

Beatrice I. Crona[1,2 ✉], Emmy Wassénius[1,2], Malin Jonell[1,2], J. Zachary Koehn[3], Rebecca Short[1], Michelle Tigchelaar[3], Tim M. Daw[1], Christopher D. Golden[4,5,6], Jessica A. Gephart[7], Edward H. Allison[8], Simon R. Bush[9], Ling Cao[10], William W. L. Cheung[11], Fabrice DeClerck[12], Jessica Fanzo[13,14], Stefan Gelcich[15,16], Avinash Kishore[17], Benjamin S. Halpern[18,19], Christina C. Hicks[20], James P. Leape[3], David C. Little[21], Fiorenza Micheli[3,22], Rosamond L. Naylor[23,24], Michael Phillips[8], Elizabeth R. Selig[3], Marco Springmann[25,26], U. Rashid Sumaila[11,27], Max Troell[2,28], Shakuntala H. Thilsted[8] & Colette C. C. Wabnitz[3,11]

Blue foods, sourced in aquatic environments, are important for the economies, livelihoods, nutritional security and cultures of people in many nations. They are often nutrient rich[1], generate lower emissions and impacts on land and water than many terrestrial meats[2], and contribute to the health[3], wellbeing and livelihoods of many rural communities[4]. The Blue Food Assessment recently evaluated nutritional, environmental, economic and justice dimensions of blue foods globally. Here we integrate these findings and translate them into four policy objectives to help realize the contributions that blue foods can make to national food systems around the world: ensuring supplies of critical nutrients, providing healthy alternatives to terrestrial meat, reducing dietary environmental footprints and safeguarding blue food contributions to nutrition, just economies and livelihoods under a changing climate. To account for how context-specific environmental, socio-economic and cultural aspects affect this contribution, we assess the relevance of each policy objective for individual countries, and examine associated co-benefits and trade-offs at national and international scales. We find that in many African and South American nations, facilitating consumption of culturally relevant blue food, especially among nutritionally vulnerable population segments, could address vitamin $B_{12}$ and omega-3 deficiencies. Meanwhile, in many global North nations, cardiovascular disease rates and large greenhouse gas footprints from ruminant meat intake could be lowered through moderate consumption of seafood with low environmental impact. The analytical framework we provide also identifies countries with high future risk, for whom climate adaptation of blue food systems will be particularly important. Overall the framework helps decision makers to assess the blue food policy objectives most relevant to their geographies, and to compare and contrast the benefits and trade-offs associated with pursuing these objectives.

Given the diverse contribution of blue foods to society, the role that they can play in the transition to healthier, more just and less environmentally harmful food systems is an important question for both public and private decision makers. Yet, blue foods have remained remarkably absent from many contemporary food system discussions and policies on both nature and nutrition-positive outcomes[5–8]. When included, their representation is often simplified and reduced to a few types of 'fish' in dietary recommendations[9] and demand projections[10]. Similarly, ocean policy often neglects blue food contributions to human nutrition and benefits to communities producing them[11,12]. Deeper appreciation and understanding of the roles blue foods can play is essential for informing policy development that can harness their

unique capacity for addressing nutritional, social and environmental food system challenges, while navigating the trade-offs of pursuing these different roles, within and across countries.

Blue foods are immensely diverse. More than 2,200 wild species are caught and more than 600 are farmed[13], with tremendous variation in associated production and processing systems and practices[2,14]. Aquatic food consumption profiles of nations are also remarkably diverse[10]. This diversity means that blue foods vary substantially in their contributions to human health, nutrition[1], jobs[15] and culture[16] and their environmental impacts[2]. Natural variations in blue food diversity and abundance are compounded by social structures that exacerbate inequities[17] across socio-economic and geographic contexts. Diversity

can bolster the resilience of blue foods to shocks[18,19], but such resilience is unevenly represented across countries at present[20].

In this diversity lies the key to understanding the geographic contexts and conditions whereby blue foods can contribute to achieving food system ambitions, such as improved nutrition, equity and lowered environmental impact—as articulated by high-level processes[21,22] and the Sustainable Development Goals of the United Nations[23].

This paper integrates the findings of an initiative to assess the multiple roles animal-sourced blue foods play in food systems around the world (the Blue Food Assessment; https://www.bluefood.earth/) and translates them into four policy objectives that could help realize the positive health, environment, resilience and equity contributions of aquatic foods worldwide. We assess the relevance of these policy objectives for individual countries, and then examine the co-benefits and trade-offs associated with policy objectives at national and international scales. In doing so, we provide a guiding framework for decision makers across public and private spheres to assess blue food policy objectives most relevant to their geographies, and compare and contrast the benefits and trade-offs associated with pursuing these objectives.

## Four ways blue foods improve food systems

The Blue Food Assessment examined the roles of blue foods in current and future food systems globally. It brought together more than 100 scholars across a wide range of disciplines to investigate the nutritional contribution of blue foods[1], current and future demand[10], and the environmental impacts of blue food production[2], as well as the vulnerability of this production to environmental stressors[24] (L.C., manuscript in preparation). It synthesized key dimensions characterizing the small-scale fisheries and aquaculture (SSFA) actors[14] who produce two-thirds of aquatic foods destined for human consumption[14,25], and evaluated injustices across the blue food system to identify policy attributes that support more equitable access to blue food benefits[17]. It also assessed the climate risks posed to nutritional, social, economic and environmental outcomes of blue food systems worldwide[20]. Finally, it explored how supporting the capabilities of actors, small and large, across the supply chains can build adaptive capacity to support a wider food system transformation (S.R.B., manuscript in preparation). This multi-perspective assessment is unique, and together with the large body of previous research helps crystallize the diverse functions blue foods play at present, and how these can be leveraged to support a food system transformation. These functions include the following.

### Sources of critical nutrients

Blue foods are rich in many essential nutrients[26]. Like other animal-source foods, blue foods can enhance bioavailability of nutrients in plant-based food sources, depending on how they are combined with other foods[27]. Where blue foods are accessible and consumed in adequate quantities, they can promote nutrition by reducing deficiencies of a range of nutrients, most notably vitamin $B_{12}$ and the omega-3 long-chain polyunsaturated fatty acids docosahexaenoic acid and eicosapentaenoic acid (DHA and EPA; hereafter, fatty acids), in which blue foods are generally rich. These are among the nutrients noted as important for human nutrition[21], showing relatively high levels of deficiency globally[1] (Extended Data Fig. 1), and blue foods are projected to contribute a global average of approximately 27% and 100% of omega-3 fatty acids and vitamin $B_{12}$, respectively, by 2030 (ref. [1]). Addressing these deficiencies is particularly important among vulnerable demographic groups, such as young children and older people, pregnant women and women of childbearing age[28,29]. Alongside other health-critical foods, blue foods can thus make essential contributions to maintaining and improving nutritional food system outcomes[30]. Capture fisheries constitute the last large wild-food resource. Failing

to sustain it will jeopardize food security in many places and it will be challenging to replace without negative environmental consequences.

### Healthy alternatives to terrestrial animal-source foods

By adding to the range of food sources associated with relative reductions in many non-communicable diseases[31–33], blue foods can help to circumvent the harmful nutrition transition observed in many countries at present (sensu ref. [34]), and contribute to reducing the overall disease burden. This may be particularly relevant in countries experiencing continued high, or growing, trends of red (particularly processed) meat intake (such as China, Argentina, Brazil, the USA and Eastern Europe)[35–37]. Cardiovascular disease is among the most commonly cited negative health effects of red meat consumption[32,33], and we use it here as an example of how countries can assess the relevance of blue food policies depending on their specific disease burden. In this context, the health-promoting role of blue foods rests on the assumption that they can displace some red meat consumption[1,10] and on the plausible health contributions (for example, of DHA and EPA from aquatic foods[38,39]), for which uncertainties persist[39–41]. Substitutability of red meat by fish has not been well documented, yet reverse substitution has been observed[30], as have large-scale adoptions of new proteins when innovations are supported with public funds, and scaled by the private sector under supportive state and international policy regimes[42]. Sixty years of increased consumption of poultry compared to beef[10] suggests that poultry and seafood can replace red meat. As blue foods are already part of the local food culture in many countries with a high level of meat consumption, they constitute a promising step away from routinized overconsumption of red meat.

### Nutrient sources with relatively low environmental footprints

Across the diversity of blue foods, many production systems already result in relatively lower environmental pressures compared to those associated with terrestrial animal-source food production[2]. Partial replacement of particularly ruminant meat with blue foods can therefore help to lower dietary environmental footprints. Unfed aquaculture systems, such as bivalves and seaweeds, typically result in low greenhouse gas (GHG), nitrogen and phosphorus emissions and require limited freshwater and land inputs. Many fed aquaculture systems perform similarly to or better than chicken production, which is often considered the most efficient terrestrial animal-source food production system[2,43]. GHG emissions for capture fisheries vary substantially[44], with small pelagic fish, cod and some inland fisheries resulting in low average emissions[45] and flounder and lobsters having high emissions[2], but all capture fisheries generally have negligible N and P emissions, and freshwater and land inputs[2,46]. Blue food production can nonetheless restructure ecological food webs and cause substantial biodiversity loss[47,48], but there is a large potential to reduce most environmental impacts associated with blue food production. Improved fisheries management, fossil-free energy and a shift to low-impact gear are key areas of interventions for capture fisheries[49]. Impacts from aquaculture could be substantially reduced by lowering feed conversion ratios (for example, through breeding programmes), shifting species focus or feed composition (for example, to deforestation-free soy, fisheries by-products, or insect meal) and improving husbandry practices[2,50,51].

### Cornerstones in cultures, diets, economies and livelihoods

In many nations, blue foods are a cornerstone of cultures, diets and economies, fulfilling critical food and nutrition security functions[4]. Blue foods are also among the most traded commodities globally, providing substantial export revenue for many nations[13] and livelihoods for 800 million people[25], indicating their critical role for employment and subsistence. However, although blue foods support the welfare (for example, through jobs and nutrient-rich blue food) of these actors[17], the wealth-generating benefits (for example, export revenues) of blue foods flow predominantly to industrial-scale firms that control global

**Table 1 | Four policy objectives delineating the role blue foods can play in addressing social, environmental and nutritional challenges of food systems in different contexts**

| Blue food policy objective | How to leverage blue food functions to address food system challenges | Conditions under which blue foods can contribute to achieving food policy objectives | Examples of co-benefits and trade-offs needing consideration |
|---|---|---|---|
| **Reducing blue-food-sensitive nutrient deficiencies** | Leverage consumption of blue food, alongside other nutrient-sensitive foods, as a means of reducing certain blue-food-sensitive nutrient deficiencies, particularly among poor or nutritionally vulnerable population segments. Our analysis centres on vitamin $B_{12}$ and omega-3, two nutrients for which blue foods are projected to contribute substantial portions of global supplies[1]. | When **nutrient insufficiency** is high and **blue foods are or can be made available**, together with other nutritious foods. | Successfully reducing blue-food-sensitive nutrient deficiencies means production portfolios must be managed strategically—developing blue food production systems with high capacity to satisfy nutritional needs with minimal environmental impact so that both environmental and nutritional co-benefits can be realized. Nutrition-sensitive trade, processing and distribution policies are also important, to avoid a scenario in which increased aquaculture production delivers foods that are not nutrient rich, affordable or accessible to those who need it. Trade-offs between economic and nutritional goals may emerge between directing national blue food production towards domestic markets versus exporting. Trade-offs between the environmental impacts of feed production and the nutritional quality of the fish produced also need to be assessed. |
| **Reducing disease burden associated with high consumption of red meat (for example, cardiovascular disease risk)** | Leverage consumption of blue food, alongside other health-promoting foods, as a means of reducing the burden of non-communicable disease related to overconsumption of red meat. Cardiovascular disease is among the most commonly cited negative health effects of overconsumption of red meat[32,33], and used here as an example for how blue foods can be leveraged to reduce specific non-communicable disease risks. | When **red meat consumption** is high, the **risk of cardiovascular disease** is high and low-environmental-impact **blue foods are or can be made available**, together with other health-promoting foods. | Blue foods vary in their environmental impacts[2]. Some have similar GHG emissions to those for poultry. By carefully considering which species are produced and traded to simultaneously minimize environmental footprints, nutritional and environmental co-benefits can be achieved. Trade-offs are otherwise likely to occur as production maximizes species that offer opportunities for efficiencies and bulk production but are not the most nutrient-dense or culturally appropriate aquatic foods. |
| **Reducing environmental footprints of food consumption and production** | Alongside overall shifts towards lower-impact diets, leverage consumption of low-impact blue foods as a means to lower GHG emissions from diets. | When **red (ruminant) meat consumption** is high and **blue foods are or can be made available**. | Reducing GHG emissions of diets through consumption of blue food can generate health co-benefits if the nutritional content of blue food groups is considered in production and trade policies. Otherwise nutritional outcomes (reduced deficiencies and disease risk) may be traded off for environmental improvement. Further health–environment co-benefits can be generated if portion sizes are limited and blue food production footprints are therefore deliberately minimized[108]. |
| **Safeguarding contributions to nutrition, just economies, livelihoods and cultures under climate change (now and in the future)** | In places where blue foods play an important role for nutrition, economies and/or employment, ensure they are climate resilient. | If **blue foods contribute substantially to national employment**, **export revenue or nutrition** and are **likely to be threatened by climate hazards**. | Safeguarding the contribution of blue foods in different settings entails reviewing production, processing and trade portfolios, as well as practices and preferences to identify relevant climate adaptation actions. However, the diversity of blue foods in terms of nutritional density, environmental impact and vulnerability to environmental stressors means that climate adaptations may present trade-offs, such as between farming species tolerant to new climate conditions but that are less nutritious. Co-benefits of climate adaptation, sustainability, health and livelihoods can be achieved if diversity in blue food supply chains is retained or enhanced. Diversity among production modes, supply chain actors and species can provide resilience to changing climatic and trade conditions, and if small-scale actors are given voice and support, it can simultaneously benefit blue-food-dependent livelihoods and contribute to nutritional security. |

Throughout the paper, sustainability refers to the need to ensure that production and consumption meet present needs without compromising those of future generations. In column three, variables used to map policy objectives to nations are in bold, and correspond to those in Supplementary Table 2. Column four provides examples of notable co-benefits and trade-offs, discussed in more detail in the main text.

supply chains[17,52,53]. This reflects inequities inherent across many food systems[54]. Small-scale actors are therefore often undervalued and marginalized in decision making, threatening livelihoods and their capacity to cope with changing environmental conditions[17,55]. Policies focusing on environmental or economic gains must therefore be attentive to risks of inadvertently undermining human wellbeing. Several environmental stressors affect blue food production, and climate change in particular will affect all aspects of aquatic food systems, from production to consumption[20]. Overall, the climate risk to a country's aquatic food system is determined not only by the climate hazards the country faces, but also by its dependence on the nutritional, economic, social and environmental benefits of aquatic foods[56], and the vulnerability to losing these benefits[20]. These future threats may compound existing challenges and exacerbate inequities, by increasing barriers to

inclusive production and trade, limiting access to blue foods, and thus restricting their nutritional contributions[14,20,57]. Supporting the diversity and resilience of SSFA[14] can help build national food system resilience to climate and other shocks[14,20,58], by providing response diversity[59]. Anticipation of how and where climate hazards will be most severe is therefore essential to help private and public actors identify appropriate actions to safeguard the contribution of blue food to the health, economies, culture and livelihoods in a way that also considers justice.

## From science to policy objectives

The potential contributions of blue foods to achieving food system ambitions depend on specific environmental, socio-economic and cultural contexts[60,61], which in turn are embedded in broader economic and political spheres[15]. We translate the blue food functions reviewed above into four policy objectives. These include leveraging consumption of blue food to: reduce vitamin $B_{12}$ and omega-3 deficiencies; reduce non-communicable disease risks related to overconsumption of red meat, particularly cardiovascular disease; and reduce GHG consumption and production footprints. A fourth policy objective centres on safeguarding blue food contributions to nutrition, just economies, livelihoods and cultures under climate change. Each objective is mapped to individual country contexts on the basis of publicly available data (Table 1), to assess the broad relevance of each policy across nations. For example, using proxy variables for insufficient nutrient intake across populations (summary exposure values of vitamin $B_{12}$ and omega-3 fatty acids), alongside blue food availability (through trade or domestic production), we identify countries for which reducing vitamin $B_{12}$ or omega-3 deficiencies among nutritionally vulnerable populations is particularly relevant (see Supplementary Table 1 for details on all variables, underlying assumptions and cutoff values). Conditions for relevance were informed by key literature and expert assessment by the interdisciplinary pool of authors. This mapping is a first step towards a more context-specific articulation of the multi-dimensional policy relevance of blue foods in food systems around the world, and could be enhanced as further data at subnational level, or for small-scale operations, become available.

## National relevance of policy objectives

The relevance of each blue food policy objective is mapped across nations globally (Fig. 1). The degree of relevance of each policy objective is evaluated by a set of rules and cutoff points (detailed in Supplementary Table 1), including a sensitivity analysis of cutoffs (Extended Data Figs. 2–6). An interactive website (https://gedb.shinyapps.io/BFA_synthesis/) presents all data and allows users to adjust cutoff points to explore the impacts of this on the relevance of each policy objective.

Fewer countries (43) were estimated to have >10% of their population at risk from inadequate intake of vitamin $B_{12}$, compared to omega-3 deficiency (89 countries), which predominantly affects African and South American nations. Many of these nations also have a high availability of blue foods, making them well positioned to address deficiencies by promoting access and facilitating consumption of culturally relevant blue food, especially among nutritionally vulnerable population segments (Fig. 1a).

Countries with red meat intake above the threshold recommended as environmentally sustainable and healthy[31], who also have a high incidence of cardiovascular disease, are primarily located in the global North, with the exception of several small-island states. In many of these countries, blue food is currently available (Fig. 1b and Supplementary Table 2). In such settings, moderate consumption of seafood with low environmental impact could be encouraged as a stepping stone away from high intake of red meat.

A substantial number of countries (124) also have a high intake of ruminant meat, contributing to high dietary GHG footprints (Fig. 1c).

Many of these countries have blue food available because they are big importers (for example, Belgium) or big producers (for example, Chile and Norway), or they both produce and import (for example, France and Denmark). Although they may export some of their blue food at present, our mapping identifies countries that, with a shift in policy or prioritization, could retain some of their domestic production for domestic consumption. This would trade off export revenue, and highlights the need to balance several policy goals, discussed below.

At present, blue foods play an important role for nutrition, livelihoods or national revenue, in a substantial number of countries (103), particularly in the global South and among Indigenous communities across the global North[4]. Combining such findings with analysis of climate hazards identifies countries with high future risk, for whom climate adaptation of blue food systems will be particularly important (Fig. 1d). We illustrate some adaptation options below.

It is important to note that for a sizeable portion of countries, certain blue food policy objectives are less relevant, according to our analysis. This does not mean that food systems in these countries are devoid of challenges, but blue foods are not a panacea and do not offer suitable means to improve food systems in all geographies at present.

## Overlapping policy relevance

For some countries, several policy objectives are relevant. Figure 2 shows the degree of overlap between policy relevance, in terms of the number of countries for which two objectives are both relevant. Some policies show a high degree of overlap, indicating possible win-wins. For example, in most (75%) of the 89 countries for which omega-3-enhancing policies are relevant, reducing environmental footprints is also a relevant objective. Similarly, for most (82%) of the 22 countries dealing with high cardiovascular disease risk, promoting blue foods over red (particularly ruminant) meat overconsumption as part of a whole-diet approach would simultaneously address health and environmental concerns.

It is noteworthy that 91% of countries with vitamin $B_{12}$ deficiencies also show high levels of omega-3 deficiency. Vitamin $B_{12}$ deficiency seems to reflect more general undernutrition of the population, whereas omega-3 deficiency (specifically DHA and EPA) is caused by low intake of blue foods. The large overlap is thus explained by most of the 42 countries whose populations are at risk of malnourishment also lacking in consumption of blue food.

In 50% of the 103 countries for which blue foods play an important role for nutrition, livelihoods or revenue, blue foods could also represent an avenue to reduce environmental footprints of ruminant meat consumption. Furthermore, in 46% of these 103 countries reducing omega-3 deficiencies is also relevant, and is similarly reflected in the 53% overlap in relevance between countries with high omega-3 deficiency and safeguarding food system contributions. In these settings, policies that can reduce certain types of malnutrition by implementing climate adaptations that ensure access to low-environmental-impact blue foods, while also securing quality jobs (that is, welfare benefits) and removing barriers to wealth-generating benefits, could therefore have the potential to generate substantial co-benefits[17,62]. Below we explore how potential co-benefits flagged by Fig. 2 can be realized.

## Harnessing diversity for co-benefits

Achieving globally agreed targets, such as zero hunger, good health, healthy aquatic and terrestrial environments, and a stable climate requires systems thinking[11,63,64]. Optimizing one policy domain often leads to both positive and negative spillover effects in several other sectors[65,66]. Addressing food system complexities that span land and sea, and/or encompass production, processing, trade and consumption, will be feasible only through systemic food policy[64,66] that

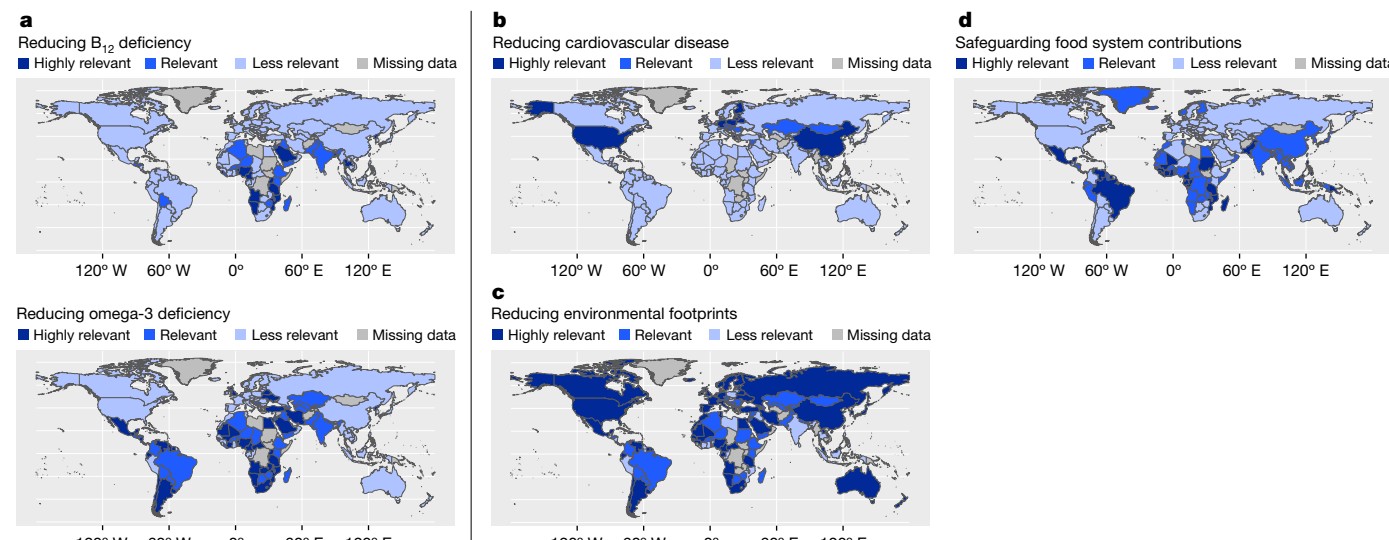

**Fig. 1 | National relevance of blue food in supporting four policy objectives.** Policy objective relevance is based on how well each nation matched the conditions for when blue foods could be expected to contribute to achieving food system ambitions (see Supplementary Table 2 for formalized inclusion criteria). **a**–**d**, The national relevance of the policies relating to reducing blue-food-sensitive deficiencies (vitamin $B_{12}$ (top) and omega-3 (bottom); **a**), reducing the burden of cardiovascular disease (**b**), reducing environmental footprints of food consumption and production (**c**) and safeguarding blue food contributions under climate change (**d**). Readers can examine the detailed objectives matching individual countries, and explore effects of different cutoffs at https://gedb.shinyapps.io/BFA_synthesis/.

identifies co-benefits between policy objectives and actions to achieve them[67]. A systemic food policy agenda can provide a clearer understanding of the potential of blue food diversity for navigating trade-offs and realizing synergies between blue food policy objectives. Below we explore opportunities for co-benefits across the policy objectives proposed above. For each, we discuss how ensuring diversity in blue food actors and blue food performance across the domains of health, nutrition, environmental impact and climate risk can help realize synergistic system-level outcomes. Overall, we argue that in any setting, shaping or maintaining food environments that make blue foods an attractive food choice is a prerequisite for achieving the multiple food system goals highlighted by this paper.

## Human health and environmental sustainability

Enabling this synergy will depend on the ability of sustainably sourced blue food to displace currently consumed foods with high environmental impact. Some aquatic foods, such as bivalves and small fish, are nutrient dense and have low environmental footprints[1,2] offering an environmentally sustainable way to address both vitamin $B_{12}$ and omega-3 deficiencies and cardiovascular disease risk. Along with cultural preferences, smell and taste, safety concerns and eating habits[68,69], price is key for determining household consumption[42]. Access and affordability are therefore prerequisites for blue foods to reduce nutrient deficiencies, cardiovascular disease risk and dietary environmental footprints[70]. However, blue food diversity means that income is a poor predictor of consumption when relying on aggregate data categories such as 'fish'[10,71]. At present, some blue foods are more expensive than other animal protein, particularly in developing contexts[63,72], but in many settings they represent affordable sources of key nutrients[10,63,73,74]. Increasing or protecting affordability will require the commoditization of low-environmental-impact blue foods through policy and regulation that promote sustainable intensification and supply chain transformation[71,75]. This can include public incentives for directing research and development investment towards specific species and production systems, and/or market incentives for value chain actors to reorient trade to low-income and nutritionally vulnerable consumers, and prioritizing increased nutrition over growth in production volumes and monetary value.

Although consumption of blue food is projected to increase (about 80% in edible weight by 2050 assuming constant prices and balance between supply and demand[10]), the resulting nutrition and environmental impacts will depend on the substitutability among blue foods and other animal-source foods in national diets. Substituting all red meats for blue foods is neither feasible nor desirable, and adding or increasing animal-source blue foods to diets of wealthy consumers, already rich in animal-source foods, would fundamentally undermine the role of blue foods in delivering healthier and less environmentally harmful dietary outcomes[2]. Fish–meat substitutability has not been widely studied, but the possibility of replacing meats with blue foods or plant-based alternatives seems to be an attractive policy option[35,72]. Strategies to achieve these goals could include combining soft policy tools such as dietary guidelines or behavioural nudging to mainstream eating and cooking blue foods[76,77], with harder regulatory interventions and economic disincentives for high-carbon-emissions food[78–81].

## Livelihoods, economies, health and environmental sustainability

Investments in blue food innovations have the potential to yield inclusive livelihoods and systems that produce nutritious, affordable and environmentally sustainable blue foods[14,75]. Such synergies again depend on which species and production modes are pursued, their variable environmental performance and nutrient density, and what barriers to access exist[1,2,82]. Production modes also vary greatly[14], from un-mechanized small-scale fisheries and farming to industrial-scale, highly specialized operations. These different production modes, and the power dynamics of supply chains developed for distribution, generally affect their contribution to equitable wealth and welfare distribution[14,17]. Policy levers are therefore needed that can improve equity by removing barriers to wealth-generating benefits. This can entail inclusive financing, infrastructure and governance that lends voice and rights to all actors and avoids displacement by competitive sectors, but also maintaining traditional access rights to nutritious blue foods; all as part of efforts to implement the human right to food[14,17,62,83]. Improving equity can also yield further benefits. For instance, increasing gender equity has been found to also improve nutritional outcomes for families[84].

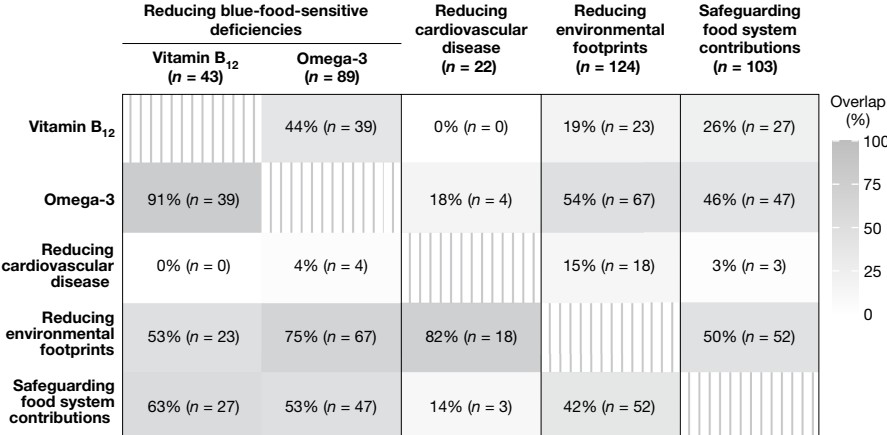

| | Reducing blue-food-sensitive deficiencies | | Reducing cardiovascular disease | Reducing environmental footprints | Safeguarding food system contributions |
| --- | --- | --- | --- | --- | --- |
| | Vitamin B$_{12}$ ($n = 43$) | Omega-3 ($n = 89$) | ($n = 22$) | ($n = 124$) | ($n = 103$) |
| Vitamin B$_{12}$ | | 44% ($n = 39$) | 0% ($n = 0$) | 19% ($n = 23$) | 26% ($n = 27$) |
| Omega-3 | 91% ($n = 39$) | | 18% ($n = 4$) | 54% ($n = 67$) | 46% ($n = 47$) |
| Reducing cardiovascular disease | 0% ($n = 0$) | 4% ($n = 4$) | | 15% ($n = 18$) | 3% ($n = 3$) |
| Reducing environmental footprints | 53% ($n = 23$) | 75% ($n = 67$) | 82% ($n = 18$) | | 50% ($n = 52$) |
| Safeguarding food system contributions | 63% ($n = 27$) | 53% ($n = 47$) | 14% ($n = 3$) | 42% ($n = 52$) | |

Overlap (%): 100, 75, 50, 25, 0

**Fig. 2 | Overlap in relevance between different policy objectives.** The numbers in parentheses in the top row represent the total number of countries for which each policy is relevant. Each cell shows the number of countries (in parentheses) for which both column- and row-heading policies are relevant, as a proportion of countries for which the column-heading policy is relevant. Relevance in this figure indicates countries categorized as 'highly relevant' or 'relevant' for a given policy.

## Climate resilience and blue food production, employment or revenue

Climate change will affect all aspects of aquatic food systems, from production to consumption, and threatens to undermine their contribution to the health, economies, culture and livelihoods of billions of people[20]. The substantial contribution that blue foods already make, particularly for livelihoods and diets, in many nations[85,86] (Fig. 1d) underscores the importance of strengthening blue food system resilience as no- or low-regret adaptation options[87]. Climate-smart production, supported through finance and adaptive governance, can reduce future disruption by promoting a multitude of different blue foods, and thus also take advantage of new opportunities that come with changing species and conditions. Examples include farming several thermally tolerant species, or introducing more flexible catch guidelines to cater for geographically extended species ranges and migration patterns of fish and fishers[88]. Addressing the current unsustainability of many fisheries by regulating harvestable quantities would simultaneously enhance stock resilience through maintenance of higher genetic diversity and thus adaptive capacity. Larger stocks are also less likely to crash when exposed to periodic shocks, such as El Niño and marine heatwaves[89]. For aquaculture, relying on a diversity of species could provide response diversity across a number of critical dimensions such as temperature, salinity or oxygen. While increasing the diversity of species to reduce climate sensitivity and increase adaptive capacity, aquaculture could also reduce the focus on fed species and promote the development of non-fed production systems[90]. Additionally, by valuing the diversity of skills and knowledge encompassed by small-scale actors and enabling their capacities to innovate and adapt to changing environmental and economic conditions, nations could further invest in the resilience of their aquatic food system (S.R.B., manuscript in preparation)[14]. Enhanced capabilities of the small-scale sector would also increase their ability to establish rights over resources, promoting more equitable forms of production and employment. Finally, disincentivizing high concentration of economic power in supply chains, characterized by the singular pursuit of efficiency gains, and mechanization at the expense of jobs[14,54] will be important to ensure that potential synergies between SSFA diversity, climate resilience and equity materialize.

## Navigating trade-offs

The complex nature of food systems, including aquatic ones, means that any action to improve performance along some dimensions will trade off performance on one or several others[65]. We identify and elaborate on three bundles of substantial trade-offs that need to be considered, but which can be navigated and minimized by making strategic use of the diversity of blue food species and production systems. We visualize an example of such trade-offs using the pursuit of either economic or nutritional blue food benefits through domestic consumption or export (Fig. 3).

### Environmental sustainability versus nutritional content of aquaculture products

In aquaculture, a pressing challenge has been to reduce reliance on wild fish for feed[50,91,92] by incorporating plant-based ingredients and recycled animal processing wastes in feeds[18,93]. However, such feeds may compromise the nutritional value of the fish produced[94] and divert produce that could be used for direct human consumption. Continued innovation to develop alternative feeds that combine lower environmental footprint with high nutritional quality will therefore be important, along with lowering feed conversion ratios[2,92], but the latter will pit improved local environmental performance against higher-quality resource requirements with consequences for sustainable and ethical resource management. Policies that promote supportive structures of governance, infrastructure and financial access to new technologies and high-quality feeds for small-scale producers will contribute to mainstreaming a move away from wild fish feeds. A regulatory environment tailored to the specific needs of blue foods, as opposed to outdated and agriculturally focused rules such as bans on use of non-ruminant processed animal proteins and genetically modified organisms, can enable rather than hinder inclusion of new protein sources. It could also avoid perverse impacts and enhance regulatory coherence for improved innovation, market access and sustainability[51].

### Domestic consumption versus export revenues

Production of blue foods for export, and allocation of fishing rights to foreign fleets, offer economic opportunities for governments, individual businesses and fishers. However, these actions can undermine domestic consumption and local livelihoods[95,96] if fishers and farmers are displaced from productive fishing or aquaculture areas[97], or suffer knock-on impacts of damaging industrial practices and overexploitation. Small-scale producers often face a tension between local needs[98] and connecting to export markets with higher profits that leave them vulnerable to global power dynamics, price fluctuations and supply chain disruptions[99]. In some cases, such as Chile, rising blue food export is associated with declining national consumption in favour

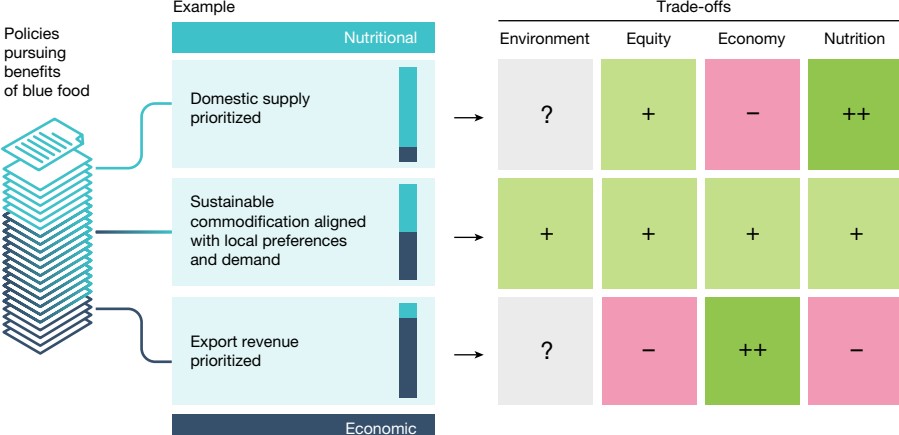

**Fig. 3 | Example of hypothetical trade-offs associated with policies pursuing economic and/or nutritional benefits of blue food.** The figure illustrates one set of trade-offs in policy outcomes that may result across the dimensions of environment, equity, economy and nutrition, depending on the degree of prioritization of either increasing domestic blue food supplies for nutritional outcome, or maximizing monetary value through exports of blue foods. The degree of emphasis placed on either policy goal is represented by the blue bars.

Likely outcomes for each dimension are represented by coloured boxes and the strength of outcome is represented by plus and minus symbols; with positive outcomes depicted in green, and negative in pink. Sustainable commodification aligned with local preferences and demand represents an example of how a balance could be struck to optimize positive environmental, inclusive, economic and nutritional outcomes. Unknown impacts, or where policy objectives are judged to not have a strong impact, are depicted in grey. E. Wikander/Azote.

of terrestrial meat[10]. As demand for blue foods rises among affluent population groups because of their contributions to health and reduced environmental footprint, prices will probably increase, exacerbating this tension. Implementing environmentally sustainable commodification will therefore be important, but must ensure that small-scale actors are not marginalized in the process (for example, ref.[100]), This requires policies that encourage collaborative practices across production scales. Cooperatives and coalitions can support complementary and synergistic production and resource access across producers[101], and inclusive jurisdictional and landscape approaches offer means to reconcile the diverse incentives and capabilities of actors in blue food production systems, while addressing ecological and geographical mismatches of current ratings and certification systems[102]. Exempting some domestic production from export is another way to secure food access but must align with local preferences, ensuring demand exists and producers are not disenfranchised[103].

### Efficiency, affordability and availability versus diversification and resilience

The global capacity to produce increased quantities of nutrient-dense, yet low-impact aquatic foods will influence the severity of trade-offs that emerge between blue food policy objectives. Commodification often offers efficiency and economies of scale, making blue foods more affordable and accessible[75], but may compromise nutrition, squeeze out small producers and processors or outcompete them in markets if measures are not in place to safeguard their livelihoods[104]. For example, large-scale production of tilapia and pangasius offers inexpensive sources of aquatic foods, but in some markets they have replaced more nutritious indigenous fish[82]. Ultimately, efficiencies must be balanced against food sovereignty and the many contributions of blue foods, distinct from their monetary value[17]. Policies to retain or enhance the diversity of blue food production modes, actors and species are essential for the capacity of nations and regions to build resilience against shocks associated with, for example, climate change[20,105], trade[19,57] or new diseases[106]. Examples include government support funds to provide financial relief for small businesses highly vulnerable to environmental and trade fluctuations[106], and improved accessibility to production-related insurance[107]. The combination of species and productions systems that provide most resilience to a changing climate

will be highly context specific, yet the species that offer opportunities for efficiencies and bulk production under a changing climate (such as tilapia with a high temperature tolerance range) may not be the most culturally appropriate or nutrient-dense aquatic foods[1]. Navigating this apparent trade-off could involve complementing bulk production of fewer species with environmentally sustainable cultivation and capture of a diversity of species that provide additional nutrition (for example, dried fish powder) and more inclusive supply chains[77].

### Policy ambitions need bold visions

The coronavirus disease 2019 pandemic has changed many aspects of our lives: how we work, travel and eat. For better or worse, it has shown that radical change is feasible in a short amount of time. For example, the pandemic highlighted how the small-scale blue foods sector was able to convey resilience and fill nutritional gaps left by interrupted global markets in some contexts, whereas in others it was left highly vulnerable[14,106]. Such shocks illustrate that backcasting is not the only, or the best, way to understand the future. Envisaging alternative futures (for example, through scenarios) may be instrumental for altering entrenched, unhealthy and unsustainable ways of producing and consuming food[80,91].

We have outlined four roles that blue foods can play now and in the future and have translated these into broad policy objectives that—if actions are developed to achieve them—could contribute to achieving articulated food system ambitions (for example, United Nations Food Systems Summit 2021). Our analysis shows that the health, environmental, economic and welfare benefits that nations derive from blue foods are diverse[60,61]. We therefore provide an analytical framework and an interactive tool (https://gedb.shinyapps.io/BFA_synthesis/) for decision makers to explore how this diversity affects the relevance of specific blue food objectives in specific contexts.

However, regardless of how environmentally sustainably produced blue foods are, the global demand for blue foods to address disease and environmental impact in one set of countries (Fig. 1b,c) may reduce the availability and affordability of blue foods for achieving improved nutritional status for vulnerable populations in another (Fig. 1a). Governments can address these tensions by regulating trade and by ensuring that diets incorporating blue foods are considered

alongside other means of achieving environmentally sustainable and healthy food system outcomes, such as various forms of more diverse and plant-rich diets[31,43,72,81]. Decisions regarding the role that blue foods can and should play for any nation's journey towards a more nutritious, equitable and less environmentally harmful food system therefore need to be grounded in local context and availability of aquatic foods, but also availability and affordability of a diversity of alternatives that are equally healthy and sustainable. Furthermore, the dietary shifts associated with the nutrition transition[34] are neither globally universal, nor inevitable. Despite their growing incomes, India and most countries in Asia–Pacific, much of the Middle East and some Latin American nations show low terrestrial meat preferences, with a higher share of protein coming from other sources, such as legumes and seafood[10]. Nations whose diets were previously constrained by low income are therefore now well placed to lead the way to sustainable and healthy eating.

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

[1]Stockholm Resilience Centre, Stockholm University, Stockholm, Sweden. [2]Global Economic Dynamics and the Biosphere, Royal Swedish Academy of Science, Stockholm, Sweden. [3]Stanford Center for Ocean Solutions, Stanford University, Stanford, CA, USA. [4]Dept. of Nutrition, Harvard T.H. Chan School of Public Health, Boston, MA, USA. [5]Dept. of Environmental Health, Harvard T.H. Chan School of Public Health, Boston, MA, USA. [6]Dept. of Global Health and Population, Harvard T.H. Chan School of Public Health, Boston, MA, USA. [7]Dept. of Environmental Science, American University, Washington, DC, USA. [8]WorldFish, Bayan Lepas, Malaysia. [9]Wageningen University and Research, Wageningen, The Netherlands. [10]School of Oceanography, Shanghai Jiao Tong University, Shanghai, China. [11]Institute for the Oceans and Fisheries, University of British Columbia, Vancouver, British Columbia, Canada. [12]EAT, Oslo, Norway. [13]Bloomberg School of Public Health, Berman Institute of Bioethics, Johns Hopkins University, Washington DC, USA. [14]Nitze School of Advanced International Studies, Johns Hopkins University, Washington, DC, USA. [15]Instituto Milenio en Socio-Ecologia Costera, Pontificia Universidad Católica de Chile, Santiago, Chile. [16]Center of Applied Ecology and Sustainability, Pontificia Universidad Católica de Chile, Santiago, Chile. [17]International Food Policy Research Institute (IFPRI), New Delhi, India. [18]National Center for Ecological Analysis and Synthesis, UC Santa Barbara, Santa Barbara, CA, USA. [19]Bren School of Environmental Science and Management, UC Santa Barbara, Santa Barbara, CA, USA. [20]Lancaster Environment Centre, Lancaster University, Lancaster, UK. [21]Institute of Aquaculture, University of Stirling, Stirling, UK. [22]Hopkins Marine Station, Oceans Department, Stanford University, Pacific Grove, CA, USA. [23]Department of Earth System Science, Stanford University, Stanford, CA, USA. [24]Center on Food Security and the Environment, Stanford University, Stanford, CA, USA. [25]Oxford Martin Programme on the Future of Food, University of Oxford, Oxford, UK. [26]Nuffield Department of Population Health, University of Oxford, Oxford, UK. [27]School of Public Policy and Global Affairs, The University of British Columbia, Vancouver, British Columbia, Canada. [28]Beijer Institute of Ecological Economics, Royal Swedish Academy of Science, Stockholm, Sweden. ✉e-mail: beatrice.crona@su.se

## Methods

### Assessing degree of policy relevance

To assess the degree of relevance of each policy for each country, we rely on theory and expert-guided typology building. Such an approach centres on classifying countries on the basis of a set of a priori assumptions about the conditions when blue food policies are relevant. The analysis has three steps.

Step one uses theory and expert assessments to build a data table of conditions that logically explain the relevance or non-relevance of each of the four policies. Conditions are variables that explain an outcome. In our analysis, these variables represent proxy variables that logically explain the relevance or non-relevance of the four policies in focus (Supplementary Table 1). The proxy variables correspond to national averages of publicly available (or published) datasets. Following best practice for related methods, such as qualitative comparative analysis[109], thresholds for inclusion (that is, cutoffs for when a policy objective is considered relevant on the basis of a country's statistic) were set on the basis of theoretical knowledge where available (Supplementary Table 1 and Extended Data Fig. 1). For many variables, however, no theoretically established cutoff existed. Thresholds were then set on the basis of natural breaks in the data after deliberation with authors of relevant expertise (see Extended Data Figs. 2–6). For all cutoff values, we provide a transparent justification for the selection and specify the type of disciplinary expertise leveraged to assess cutoffs for each variable (Supplementary Table 1).

A second step involves developing Boolean logic solution formulae that allow us to classify countries in relation to the outcome variable 'degree of policy relevance' (highly relevant, relevant, less relevant and missing data). We use a crisp set methodology to define which countries are relevant for a particular policy (see Supplementary Information for elaborated justification). Crisp sets assign cases (countries) a binary value for each variable. The binary value is based on whether the data for the country fall above or below the pre-determined cutoff (Supplementary Table 1). Logic solution formulae specify the combination of binary conditions that results in a given level of policy relevance (Supplementary Table 2), and these solution formulae are based on expert judgement and logic (Supplementary Table 1). The combination of AND and OR statements in the solution formulae highlight the distinction between necessary and sufficient conditions, akin to how these are conceptualized in qualitative comparative analysis[110].

All datasets used as input into the Boolean analysis (referenced in Supplementary Table 1) are freely available through peer-reviewed publications or publicly available databases. These include: ref. [1]; Food and Agriculture Organization of the United Nations (FAO) Fishery and Aquaculture Statistics, Global capture production 1950–2019 (FishstatJ), available at https://www.fao.org/fishery/en/statistics (ref. [13]); global expanded nutrient supply model, available at https://dataverse.harvard.edu/dataset.xhtml?persistentId=doi:10.7910/DVN/5LC3SI ; World Health Organization, Global health estimates: leading causes of DALYs, available at https://www.who.int/data/gho/data/themes/mortality-and-global-health-estimates/global-health-estimates-leading-causes-of-dalys; (ref. [15]); FAO Yearbook, Fishery and Aquaculture Statistics 2018, available at https://www.fao.org/fishery/en/publications/269665; ILOSTAT labour statistics (2020), available at https://ilostat.ilo.org/; World Development Indicators (World Bank) DataBank (2012), available at https://databank.worldbank.org/reports.aspx?source=world-development-indicators (ref. [37]); FAOSTAT Food Balances, available at http://www.fao.org/faostat/en/#data/FBS; the variable 'hazard by system' in the web application uses data presented in the extended data for ref. [20], available at https://doi.org/10.1038/s43016-021-00368-9.

One key reason for choosing crisp set methodology was that we wanted to maximize the ease of interpretation and potential use. Crisp sets arguably retain less information richness than fuzzy sets (for which membership of cases is not binary but assigned as degrees of membership to different categories). However, although partial set membership allows for more information from the underlying data to be maintained, it is also likely to result in situations of partial relevance in the outcome variable (degree of policy relevance). In other words, one could easily end up in a situation in which a country is classified as 33% relevant. This would be exceedingly hard for readers to interpret and act on. In other words, crisp sets were chosen in order for countries to receive a clear classification of relevance (highly relevant, relevant, less relevant and missing data) in our analysis. Another reason for opting for crisp sets is the above noted lack of scientific consensus to guide the exact cutoffs for all variables assessing the conditions. Fuzzy set analysis requires several such decisions to be made as each variable is divided into a minimum of three sets (as opposed to a case simply being in the set = 1, or out = 0), and would have thus increased the uncertainty of the analysis. We recognize that even with our analysis some cutoffs could be up for discussion. We therefore invite readers to explore different threshold values and see the change in outcome in the web-based tool available at https://gedb.shinyapps.io/BFA_synthesis/. This tool also means that, as scientific evidence for a particular cutoff value becomes available or updated, this information can easily be applied to revise the classification of nations.

The third step of our analytical approach involves matching the set configurations in the data table (step 1) to the Boolean logic solution formulae designed in step 2, to assign each case (country) to the outcome variable 'degree of policy relevance' (Supplementary Table 2). This outcome variable (for each policy objective) forms the results presented in Fig. 1, and forms the basis of the overlap analysis presented in Fig. 2.

### Sensitivity analysis

To assess the sensitivity of our results to variations in thresholds, we opted for a one-at-a-time sensitivity approach[111] (Extended Data Figs. 2–6) owing to the simplicity of Boolean rules used in this analysis. This was combined with a visual examination of the underlying distribution of each variable used in the analysis, in relation to our set threshold (Extended Data Fig. 1). The typology classification model was re-run changing one variable threshold at a time, leaving all else constant, and assessing the change in the number of countries in each outcome category ('highly relevant', 'relevant' and 'less relevant'). The threshold was varied across the full range of the variable. The sensitivity analysis highlights to what degree the underlying distribution of the data (Extended Data Fig. 1), as well as the Boolean logic on which the classification model is based (Supplementary Table 2), influences the sensitivity of the threshold.

### Assessing overlap in policy relevance

To assess overlap in policy objective relevance among nations, we conducted pairwise comparison of policy objectives. We calculated the percentage of the set of countries to whom a specific policy was deemed relevant, and that was also deemed relevant for a second policy objective. Nations classified as 'highly relevant' or 'relevant' were combined for the purpose of this analysis, and countries with missing data for either policy were not considered.

### Reporting summary

Further information on research design is available in the Nature Portfolio Reporting Summary linked to this article.

## Data availability

All data generated and analysed during the study are available in the Stockholm University Library Dataverse (https://doi.org/10.7910/DVN/ILA0XI).

## Code availability

R code used for the Boolean analysis and sensitivity analysis, as well as for producing the figures and web application, is available at https://github.com/emmywas/BFA_Policy_analysis. All coding was carried out in R (version 4.2.0).

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

**Acknowledgements** This paper is part of the Blue Food Assessment (https://www.bluefood.earth), a comprehensive examination of the role of aquatic foods in building healthy, sustainable and equitable food systems. The assessment was supported by the Builders Initiative, the MAVA Foundation, the Oak Foundation and the Walton Family Foundation. We thank all scientific leaders of the Blue Food Assessment for their intellectual input on this paper, as well as S. Maniatakou and G. Parlato for research assistance. B.C. also acknowledges the generous support of the Erling Persson Family Foundation.

**Author contributions** B.C. conceptualized the study, with substantial methodological and design input from T.D., E.W., M. Tigchelaar, M.J., R.S. and J.Z.K. as well as C.D.G. and J.A.G. Data acquisition and compilation was conducted by E.W., M. Tigchelaar, M.J., R.S. and J.Z.K., and analysis was conducted by B.C. and E.W., with input from T.D. B.C. drafted the original manuscript and all co-authors reviewed and revised the writing and interpretation of findings.

**Funding** Open access funding provided by Stockholm University.

**Competing interests** R.S. sits on the board of Oceana, and C.D.G., R.L.N. and J.A.G. serve as scientific advisers to the same organization. S.R.B. has unpaid advisory roles on the International Advisory Board for Aquaculture Investments of the Sustainable Trade Initiative (IDH), Utrecht, The Netherlands; and on the Standards Oversight Committee of the Global Seafood Alliance, United States; and is part of the Seafood Watch Aquaculture Multi Stakeholder Group at Monterey Bay Aquarium, CA, USA. B.C., M. Troell, E.R.S. and C.C.C.W. provide occasional voluntary and unpaid scientific support to the Seafood Business for Ocean Stewardship initiative (https://seabos.org/). None of the non-academic actors mentioned has had any input to the study design, analysis, interpretation of data or conclusions drawn.

### Additional information

**Correspondence and requests for materials** should be addressed to Beatrice I. Crona.

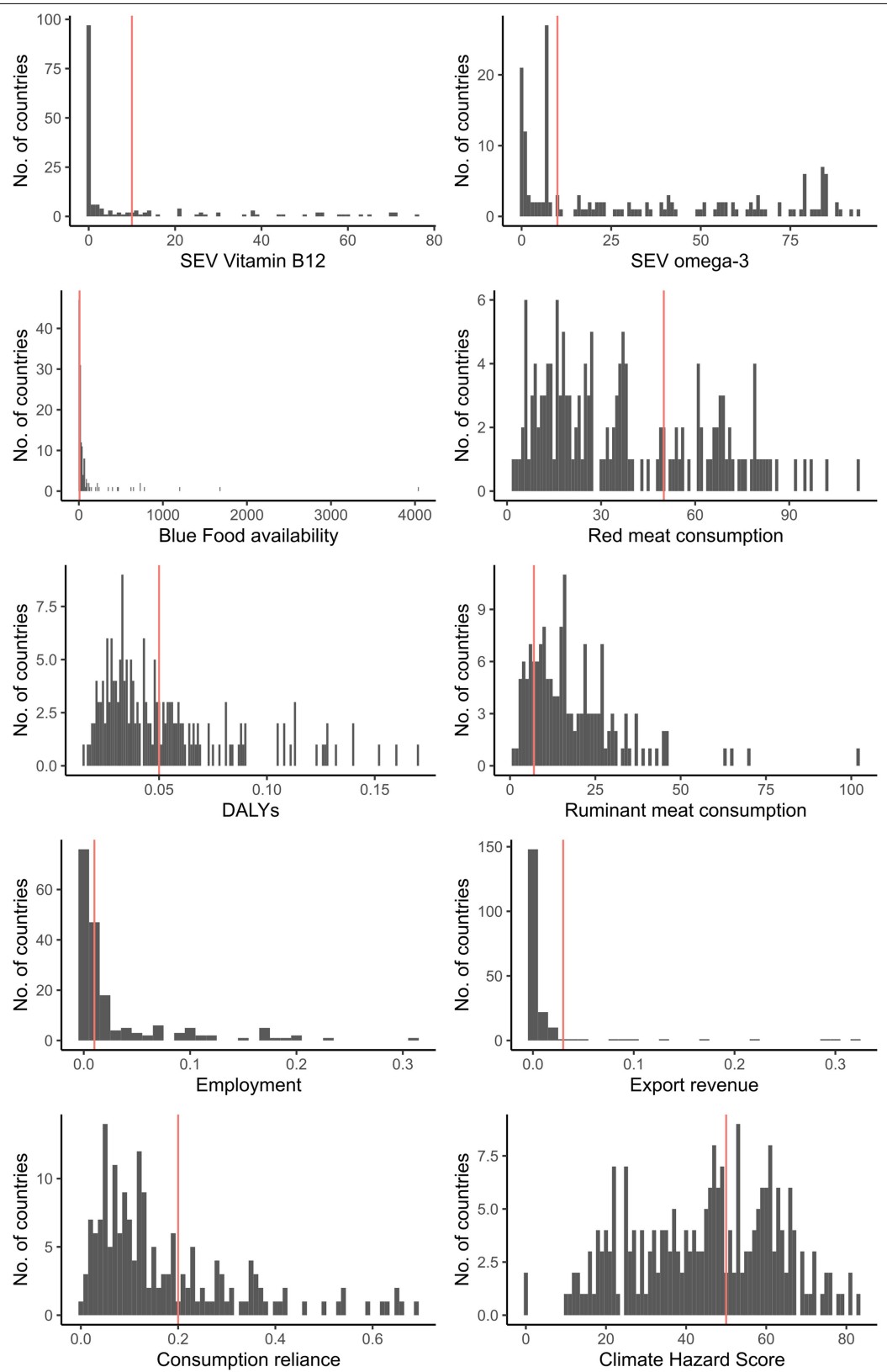

**Extended Data Fig. 1 | Underlying distribution of variables.** Red line indicates selected cut-offs used in our analysis. In cases where many countries have data close to the cutoff – a change in threshold value will greatly impact the outcomes, thus explaining some of the results of the sensitivity analyses (see Extended Data Figs. 2–6).

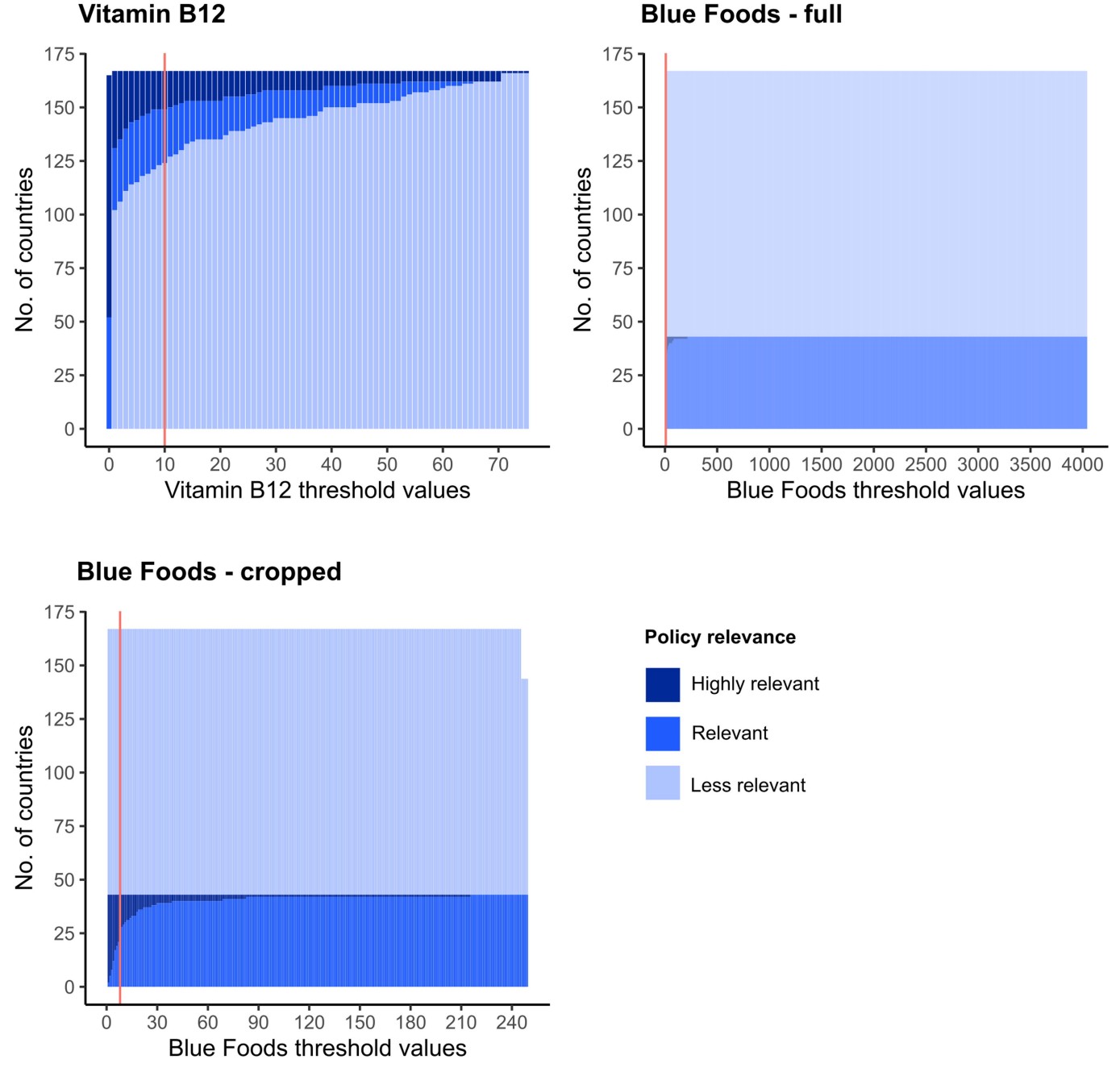

**Extended Data Fig. 2 | Sensitivity analysis of policy "Reducing blue food sensitive nutrient deficiencies" – for vitamin B12.** Shows number of countries in each category of policy relevance (highly relevant, relevant, less relevant), under all possible values of the threshold. Blue food variable is shown both in its full extent and in a cropped version to highlight the variability around the selected threshold. Red vertical line indicates selected threshold in analysis.

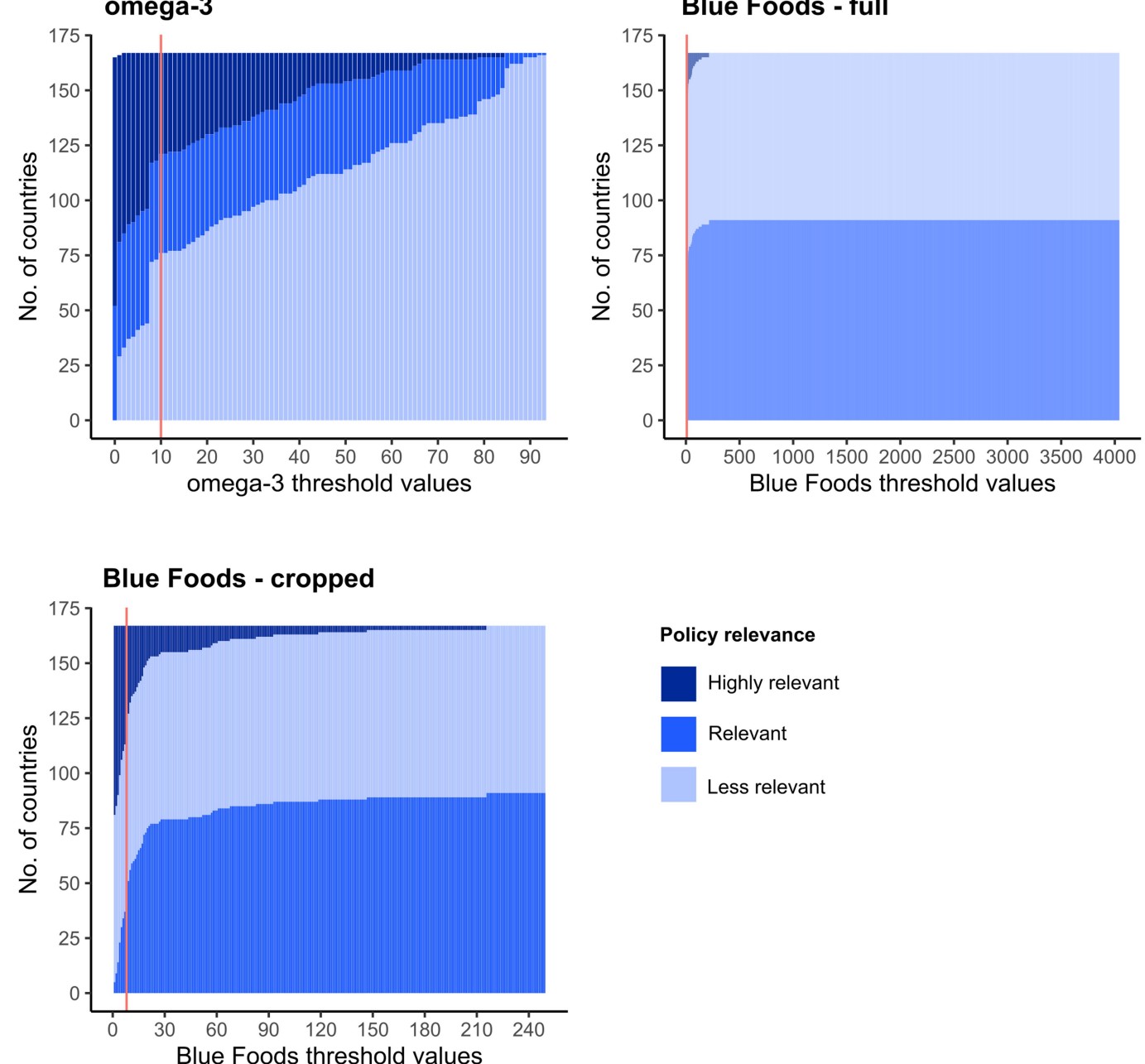

**Extended Data Fig. 3 | Sensitivity analysis of policy "Reducing blue food sensitive nutrient deficiencies" – for omega-3.** Shows number of countries in each category of policy relevance (highly relevant, relevant, less relevant), under all possible values of the threshold. Blue food variable is shown both in its full extent and in a cropped version to highlight the variability around the selected threshold. Red vertical line indicates selected threshold in analysis.

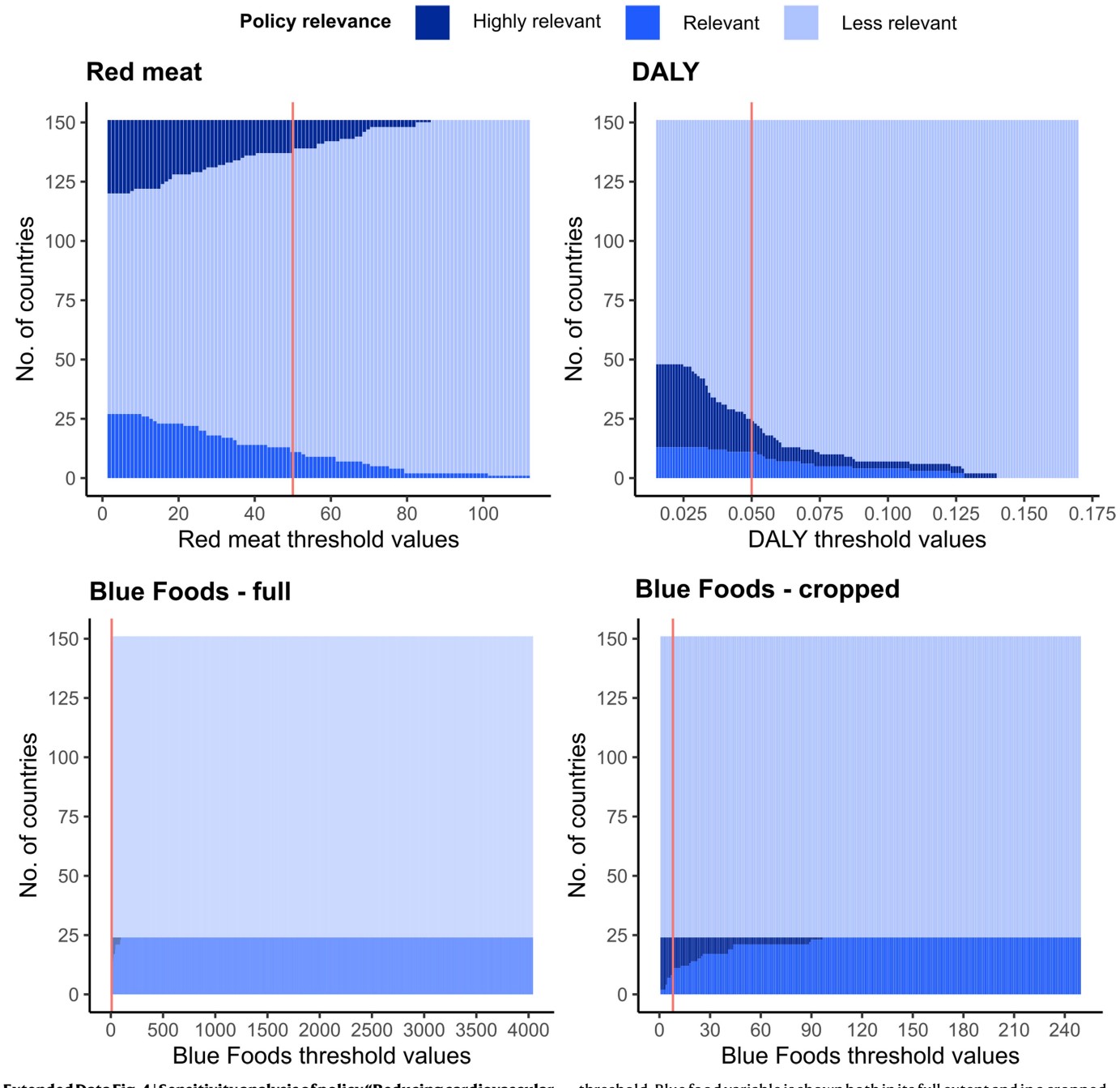

**Extended Data Fig. 4 | Sensitivity analysis of policy "Reducing cardiovascular disease risk".** Shows number of countries in each category of policy relevance (highly relevant, relevant, less relevant), under all possible values of the threshold. Blue food variable is shown both in its full extent and in a cropped version to highlight the variability around the selected threshold. Red vertical line indicates selected threshold in analysis.

## Ruminant meat

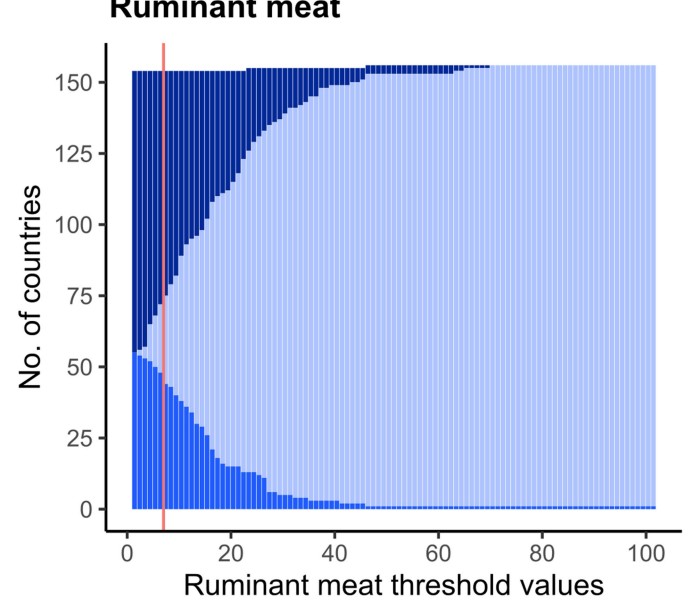

**Policy relevance**

- Highly relevant
- Relevant
- Less relevant

## Blue Foods - full

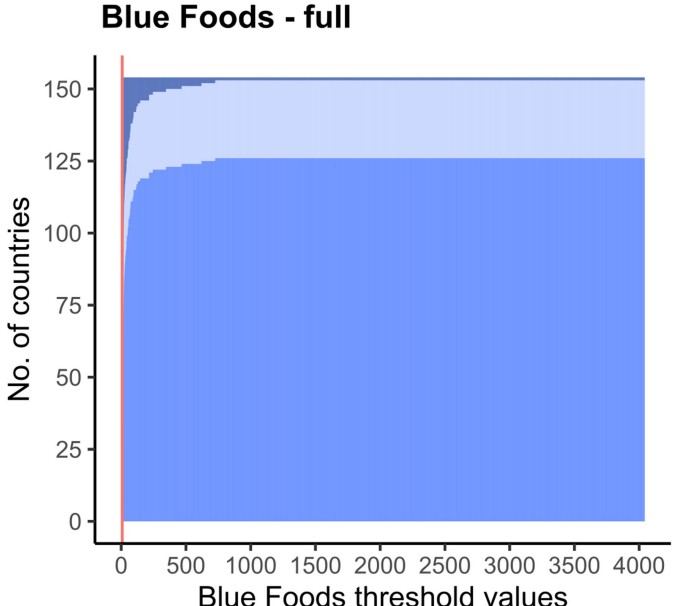

## Blue Foods - cropped

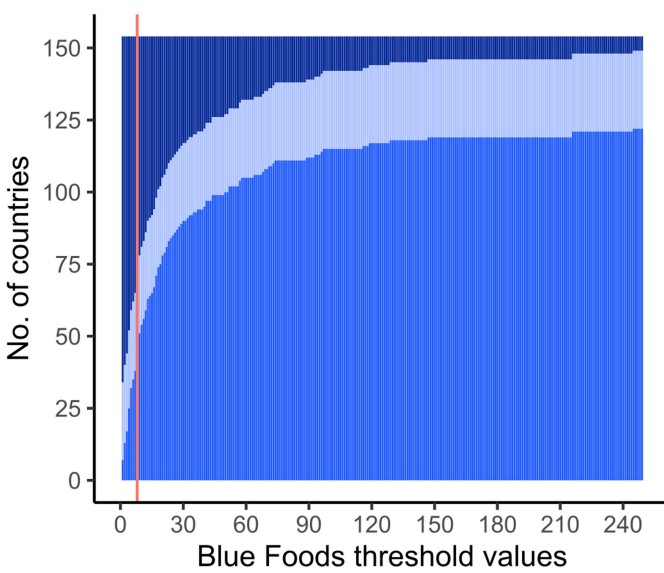

**Extended Data Fig. 5 | Sensitivity analysis of policy "Reducing environmental footprints of food consumption and production".** Shows number of countries in each category of policy relevance (highly relevant, relevant, less relevant), under all possible values of the threshold. Blue food variable is shown both in its full extent and in a cropped version to highlight the variability around the selected threshold. Red vertical line indicates selected threshold in analysis.

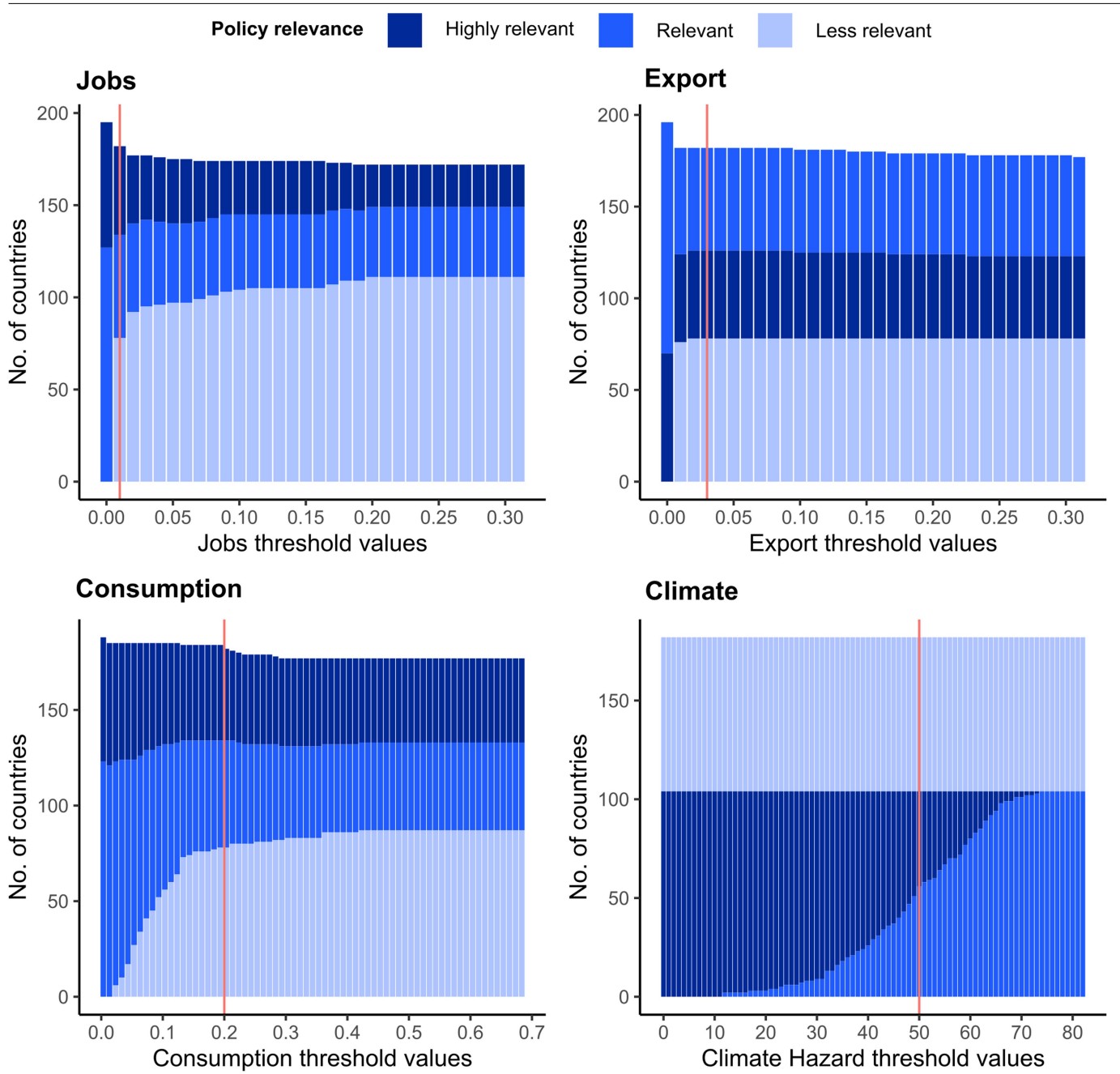

**Extended Data Fig. 6 | Sensitivity analysis of policy "Safeguarding food system contributions under climate change".** Shows number of countries in each category of policy relevance (highly relevant, relevant, less relevant), under all possible values of the threshold. Red vertical line indicates selected threshold in analysis.

# Reporting Summary

## Statistics

For all statistical analyses, confirm that the following items are present in the figure legend, table legend, main text, or Methods section.

| n/a | Confirmed | |
|---|---|---|
| ☐ | ☒ | The exact sample size (*n*) for each experimental group/condition, given as a discrete number and unit of measurement |
| ☒ | ☐ | A statement on whether measurements were taken from distinct samples or whether the same sample was measured repeatedly |
| ☐ | ☒ | The statistical test(s) used AND whether they are one- or two-sided *Only common tests should be described solely by name; describe more complex techniques in the Methods section.* |
| ☒ | ☐ | A description of all covariates tested |
| ☐ | ☒ | A description of any assumptions or corrections, such as tests of normality and adjustment for multiple comparisons |
| ☒ | ☐ | A full description of the statistical parameters including central tendency (e.g. means) or other basic estimates (e.g. regression coefficient) AND variation (e.g. standard deviation) or associated estimates of uncertainty (e.g. confidence intervals) |
| ☒ | ☐ | For null hypothesis testing, the test statistic (e.g. *F*, *t*, *r*) with confidence intervals, effect sizes, degrees of freedom and *P* value noted *Give P values as exact values whenever suitable.* |
| ☒ | ☐ | For Bayesian analysis, information on the choice of priors and Markov chain Monte Carlo settings |
| ☒ | ☐ | For hierarchical and complex designs, identification of the appropriate level for tests and full reporting of outcomes |
| ☒ | ☐ | Estimates of effect sizes (e.g. Cohen's *d*, Pearson's *r*), indicating how they were calculated |

*Our web collection on statistics for biologists contains articles on many of the points above.*

## Software and code

Policy information about availability of computer code

| Data collection | no software was used to collect data, data was manually extracted from existing datasets |
|---|---|
| Data analysis | all coding was done in R (version 4.2.0), and all code to replicate the analysis is available at: https://github.com/emmywas/BFA_Policy_analysis |

For manuscripts utilizing custom algorithms or software that are central to the research but not yet described in published literature, software must be made available to editors and reviewers. We strongly encourage code deposition in a community repository (e.g. GitHub). See the Nature Portfolio guidelines for submitting code & software for further information.

## Data

Policy information about availability of data

All manuscripts must include a data availability statement. This statement should provide the following information, where applicable:
- Accession codes, unique identifiers, or web links for publicly available datasets
- A description of any restrictions on data availability
- For clinical datasets or third party data, please ensure that the statement adheres to our policy

All datasets used as input into the Boolean analysis (referenced in Table S1) are freely available via peer-reviewed publications or publically available databases. These include;
Golden, C. D. et al. Aquatic Foods for Nourishing Nations. Nature 598, 315-320 (2021).
FAO Fishery and Aquaculture Statistics. Global capture production 1950 - 2019 (FishstatJ). Available at: https://www.fao.org/fishery/en/statistics

Global expanded nutrient supply (GENuS) model. Available at: https://dataverse.harvard.edu/dataset.xhtml?persistentId=doi:10.7910/DVN/5LC3SI

WHO. Global health estimates: Leading causes of DALYs. Available at: https://www.who.int/data/gho/data/themes/mortality-and-global-health-estimates/global-health-estimates-leading-causes-of-dalys

Teh, L. C. L. & Sumaila, U. R. Contribution of marine fisheries to worldwide employment. Fish Fish. 14, 77–88 (2013).

FAO Yearbook. Fishery and Aquaculture Statistics 2018. Available at: https://www.fao.org/fishery/en/publications/269665

ILOSTAT labour statistics. (2020). Available at: https://ilostat.ilo.org/

World Development Indicators (World Bank) DataBank (2012). Available at: https://databank.worldbank.org/reports.aspx?source=world-development-indicators

FAOSTAT Food Balances. Available at: http://www.fao.org/faostat/en/#data/FBS

The variable 'hazard by system' in the web application uses data presented in extended data for article Tigchelaar, M., Cheung, W.W.L., Mohammed, E.Y. et al. Compound climate risks threaten aquatic food system benefits. Nat Food 2, 673–682 (2021). Available at: https://doi.org/10.1038/s43016-021-00368-9

All data generated and analysed during the study are available in the Stockholm University Library Dataverse (https://doi.org/10.7910/DVN/ILA0XI).

# Human research participants

Policy information about studies involving human research participants and Sex and Gender in Research.

**Reporting on sex and gender**
*Use the terms sex (biological attribute) and gender (shaped by social and cultural circumstances) carefully in order to avoid confusing both terms. Indicate if findings apply to only one sex or gender; describe whether sex and gender were considered in study design whether sex and/or gender was determined based on self-reporting or assigned and methods used. Provide in the source data disaggregated sex and gender data where this information has been collected, and consent has been obtained for sharing of individual-level data; provide overall numbers in this Reporting Summary. Please state if this information has not been collected. Report sex- and gender-based analyses where performed, justify reasons for lack of sex- and gender-based analysis.*

**Population characteristics**
*Describe the covariate-relevant population characteristics of the human research participants (e.g. age, genotypic information, past and current diagnosis and treatment categories). If you filled out the behavioural & social sciences study design questions and have nothing to add here, write "See above."*

**Recruitment**
*Describe how participants were recruited. Outline any potential self-selection bias or other biases that may be present and how these are likely to impact results.*

**Ethics oversight**
*Identify the organization(s) that approved the study protocol.*

Note that full information on the approval of the study protocol must also be provided in the manuscript.

# Field-specific reporting

Please select the one below that is the best fit for your research. If you are not sure, read the appropriate sections before making your selection.

☐ Life sciences  ☐ Behavioural & social sciences  ☐ Ecological, evolutionary & environmental sciences

For a reference copy of the document with all sections, see nature.com/documents/nr-reporting-summary-flat.pdf

# Life sciences study design

All studies must disclose on these points even when the disclosure is negative.

**Sample size**
*Describe how sample size was determined, detailing any statistical methods used to predetermine sample size OR if no sample-size calculation was performed, describe how sample sizes were chosen and provide a rationale for why these sample sizes are sufficient.*

**Data exclusions**
*Describe any data exclusions. If no data were excluded from the analyses, state so OR if data were excluded, describe the exclusions and the rationale behind them, indicating whether exclusion criteria were pre-established.*

**Replication**
*Describe the measures taken to verify the reproducibility of the experimental findings. If all attempts at replication were successful, confirm this OR if there are any findings that were not replicated or cannot be reproduced, note this and describe why.*

**Randomization**
*Describe how samples/organisms/participants were allocated into experimental groups. If allocation was not random, describe how covariates were controlled OR if this is not relevant to your study, explain why.*

**Blinding**
*Describe whether the investigators were blinded to group allocation during data collection and/or analysis. If blinding was not possible, describe why OR explain why blinding was not relevant to your study.*

# Behavioural & social sciences study design

All studies must disclose on these points even when the disclosure is negative.

| | |
|---|---|
| Study description | *Briefly describe the study type including whether data are quantitative, qualitative, or mixed-methods (e.g. qualitative cross-sectional, quantitative experimental, mixed-methods case study).* |
| Research sample | *State the research sample (e.g. Harvard university undergraduates, villagers in rural India) and provide relevant demographic information (e.g. age, sex) and indicate whether the sample is representative. Provide a rationale for the study sample chosen. For studies involving existing datasets, please describe the dataset and source.* |
| Sampling strategy | *Describe the sampling procedure (e.g. random, snowball, stratified, convenience). Describe the statistical methods that were used to predetermine sample size OR if no sample-size calculation was performed, describe how sample sizes were chosen and provide a rationale for why these sample sizes are sufficient. For qualitative data, please indicate whether data saturation was considered, and what criteria were used to decide that no further sampling was needed.* |
| Data collection | *Provide details about the data collection procedure, including the instruments or devices used to record the data (e.g. pen and paper, computer, eye tracker, video or audio equipment) whether anyone was present besides the participant(s) and the researcher, and whether the researcher was blind to experimental condition and/or the study hypothesis during data collection.* |
| Timing | *Indicate the start and stop dates of data collection. If there is a gap between collection periods, state the dates for each sample cohort.* |
| Data exclusions | *If no data were excluded from the analyses, state so OR if data were excluded, provide the exact number of exclusions and the rationale behind them, indicating whether exclusion criteria were pre-established.* |
| Non-participation | *State how many participants dropped out/declined participation and the reason(s) given OR provide response rate OR state that no participants dropped out/declined participation.* |
| Randomization | *If participants were not allocated into experimental groups, state so OR describe how participants were allocated to groups, and if allocation was not random, describe how covariates were controlled.* |

# Ecological, evolutionary & environmental sciences study design

All studies must disclose on these points even when the disclosure is negative.

| | |
|---|---|
| Study description | *Briefly describe the study. For quantitative data include treatment factors and interactions, design structure (e.g. factorial, nested, hierarchical), nature and number of experimental units and replicates.* |
| Research sample | *Describe the research sample (e.g. a group of tagged Passer domesticus, all Stenocereus thurberi within Organ Pipe Cactus National Monument), and provide a rationale for the sample choice. When relevant, describe the organism taxa, source, sex, age range and any manipulations. State what population the sample is meant to represent when applicable. For studies involving existing datasets, describe the data and its source.* |
| Sampling strategy | *Note the sampling procedure. Describe the statistical methods that were used to predetermine sample size OR if no sample-size calculation was performed, describe how sample sizes were chosen and provide a rationale for why these sample sizes are sufficient.* |
| Data collection | *Describe the data collection procedure, including who recorded the data and how.* |
| Timing and spatial scale | *Indicate the start and stop dates of data collection, noting the frequency and periodicity of sampling and providing a rationale for these choices. If there is a gap between collection periods, state the dates for each sample cohort. Specify the spatial scale from which the data are taken* |
| Data exclusions | *If no data were excluded from the analyses, state so OR if data were excluded, describe the exclusions and the rationale behind them, indicating whether exclusion criteria were pre-established.* |
| Reproducibility | *Describe the measures taken to verify the reproducibility of experimental findings. For each experiment, note whether any attempts to repeat the experiment failed OR state that all attempts to repeat the experiment were successful.* |
| Randomization | *Describe how samples/organisms/participants were allocated into groups. If allocation was not random, describe how covariates were controlled. If this is not relevant to your study, explain why.* |
| Blinding | *Describe the extent of blinding used during data acquisition and analysis. If blinding was not possible, describe why OR explain why blinding was not relevant to your study.* |

Did the study involve field work? ☐ Yes ☐ No

## Field work, collection and transport

| | |
|---|---|
| Field conditions | *Describe the study conditions for field work, providing relevant parameters (e.g. temperature, rainfall).* |
| Location | *State the location of the sampling or experiment, providing relevant parameters (e.g. latitude and longitude, elevation, water depth).* |
| Access & import/export | *Describe the efforts you have made to access habitats and to collect and import/export your samples in a responsible manner and in compliance with local, national and international laws, noting any permits that were obtained (give the name of the issuing authority, the date of issue, and any identifying information).* |
| Disturbance | *Describe any disturbance caused by the study and how it was minimized.* |

# Reporting for specific materials, systems and methods

We require information from authors about some types of materials, experimental systems and methods used in many studies. Here, indicate whether each material, system or method listed is relevant to your study. If you are not sure if a list item applies to your research, read the appropriate section before selecting a response.

### Materials & experimental systems

| n/a | Involved in the study |
|---|---|
| ☐ | ☐ Antibodies |
| ☐ | ☐ Eukaryotic cell lines |
| ☐ | ☐ Palaeontology and archaeology |
| ☐ | ☐ Animals and other organisms |
| ☐ | ☐ Clinical data |
| ☐ | ☐ Dual use research of concern |

### Methods

| n/a | Involved in the study |
|---|---|
| ☐ | ☐ ChIP-seq |
| ☐ | ☐ Flow cytometry |
| ☐ | ☐ MRI-based neuroimaging |

## Antibodies

| | |
|---|---|
| Antibodies used | *Describe all antibodies used in the study; as applicable, provide supplier name, catalog number, clone name, and lot number.* |
| Validation | *Describe the validation of each primary antibody for the species and application, noting any validation statements on the manufacturer's website, relevant citations, antibody profiles in online databases, or data provided in the manuscript.* |

## Eukaryotic cell lines

Policy information about cell lines and Sex and Gender in Research

| | |
|---|---|
| Cell line source(s) | *State the source of each cell line used and the sex of all primary cell lines and cells derived from human participants or vertebrate models.* |
| Authentication | *Describe the authentication procedures for each cell line used OR declare that none of the cell lines used were authenticated.* |
| Mycoplasma contamination | *Confirm that all cell lines tested negative for mycoplasma contamination OR describe the results of the testing for mycoplasma contamination OR declare that the cell lines were not tested for mycoplasma contamination.* |
| Commonly misidentified lines (See ICLAC register) | *Name any commonly misidentified cell lines used in the study and provide a rationale for their use.* |

## Palaeontology and Archaeology

| | |
|---|---|
| Specimen provenance | *Provide provenance information for specimens and describe permits that were obtained for the work (including the name of the issuing authority, the date of issue, and any identifying information). Permits should encompass collection and, where applicable, export.* |
| Specimen deposition | *Indicate where the specimens have been deposited to permit free access by other researchers.* |

| Dating methods | *If new dates are provided, describe how they were obtained (e.g. collection, storage, sample pretreatment and measurement), where they were obtained (i.e. lab name), the calibration program and the protocol for quality assurance OR state that no new dates are provided.* |

☐ Tick this box to confirm that the raw and calibrated dates are available in the paper or in Supplementary Information.

| Ethics oversight | *Identify the organization(s) that approved or provided guidance on the study protocol, OR state that no ethical approval or guidance was required and explain why not.* |

Note that full information on the approval of the study protocol must also be provided in the manuscript.

## Animals and other research organisms

Policy information about studies involving animals; ARRIVE guidelines recommended for reporting animal research, and Sex and Gender in Research

| Laboratory animals | *For laboratory animals, report species, strain and age OR state that the study did not involve laboratory animals.* |
| Wild animals | *Provide details on animals observed in or captured in the field; report species and age where possible. Describe how animals were caught and transported and what happened to captive animals after the study (if killed, explain why and describe method; if released, say where and when) OR state that the study did not involve wild animals.* |
| Reporting on sex | *Indicate if findings apply to only one sex; describe whether sex was considered in study design, methods used for assigning sex. Provide data disaggregated for sex where this information has been collected in the source data as appropriate; provide overall numbers in this Reporting Summary. Please state if this information has not been collected. Report sex-based analyses where performed, justify reasons for lack of sex-based analysis.* |
| Field-collected samples | *For laboratory work with field-collected samples, describe all relevant parameters such as housing, maintenance, temperature, photoperiod and end-of-experiment protocol OR state that the study did not involve samples collected from the field.* |
| Ethics oversight | *Identify the organization(s) that approved or provided guidance on the study protocol, OR state that no ethical approval or guidance was required and explain why not.* |

Note that full information on the approval of the study protocol must also be provided in the manuscript.

## Clinical data

Policy information about clinical studies

All manuscripts should comply with the ICMJE guidelines for publication of clinical research and a completed CONSORT checklist must be included with all submissions.

| Clinical trial registration | *Provide the trial registration number from ClinicalTrials.gov or an equivalent agency.* |
| Study protocol | *Note where the full trial protocol can be accessed OR if not available, explain why.* |
| Data collection | *Describe the settings and locales of data collection, noting the time periods of recruitment and data collection.* |
| Outcomes | *Describe how you pre-defined primary and secondary outcome measures and how you assessed these measures.* |

## Dual use research of concern

Policy information about dual use research of concern

### Hazards

Could the accidental, deliberate or reckless misuse of agents or technologies generated in the work, or the application of information presented in the manuscript, pose a threat to:

No | Yes

☐ ☐ Public health

☐ ☐ National security

☐ ☐ Crops and/or livestock

☐ ☐ Ecosystems

☐ ☐ Any other significant area

## Experiments of concern

Does the work involve any of these experiments of concern:

No Yes

☐ ☐ Demonstrate how to render a vaccine ineffective

☐ ☐ Confer resistance to therapeutically useful antibiotics or antiviral agents

☐ ☐ Enhance the virulence of a pathogen or render a nonpathogen virulent

☐ ☐ Increase transmissibility of a pathogen

☐ ☐ Alter the host range of a pathogen

☐ ☐ Enable evasion of diagnostic/detection modalities

☐ ☐ Enable the weaponization of a biological agent or toxin

☐ ☐ Any other potentially harmful combination of experiments and agents

# ChIP-seq

## Data deposition

☐ Confirm that both raw and final processed data have been deposited in a public database such as GEO.

☐ Confirm that you have deposited or provided access to graph files (e.g. BED files) for the called peaks.

| Data access links | For "Initial submission" or "Revised version" documents, provide reviewer access links.  For your "Final submission" document, provide a link to the deposited data. |
| --- | --- |
| *May remain private before publication.* | |
| Files in database submission | Provide a list of all files available in the database submission. |
| Genome browser session (e.g. UCSC) | Provide a link to an anonymized genome browser session for "Initial submission" and "Revised version" documents only, to enable peer review.  Write "no longer applicable" for "Final submission" documents. |

## Methodology

| Replicates | Describe the experimental replicates, specifying number, type and replicate agreement. |
| --- | --- |
| Sequencing depth | Describe the sequencing depth for each experiment, providing the total number of reads, uniquely mapped reads, length of reads and whether they were paired- or single-end. |
| Antibodies | Describe the antibodies used for the ChIP-seq experiments; as applicable, provide supplier name, catalog number, clone name, and lot number. |
| Peak calling parameters | Specify the command line program and parameters used for read mapping and peak calling, including the ChIP, control and index files used. |
| Data quality | Describe the methods used to ensure data quality in full detail, including how many peaks are at FDR 5% and above 5-fold enrichment. |
| Software | Describe the software used to collect and analyze the ChIP-seq data. For custom code that has been deposited into a community repository, provide accession details. |

# Flow Cytometry

## Plots

Confirm that:

☐ The axis labels state the marker and fluorochrome used (e.g. CD4-FITC).

☐ The axis scales are clearly visible. Include numbers along axes only for bottom left plot of group (a 'group' is an analysis of identical markers).

☐ All plots are contour plots with outliers or pseudocolor plots.

☐ A numerical value for number of cells or percentage (with statistics) is provided.

## Methodology

| Sample preparation | Describe the sample preparation, detailing the biological source of the cells and any tissue processing steps used. |
| --- | --- |
| Instrument | Identify the instrument used for data collection, specifying make and model number. |

| Software | *Describe the software used to collect and analyze the flow cytometry data. For custom code that has been deposited into a community repository, provide accession details.* |
|---|---|
| Cell population abundance | *Describe the abundance of the relevant cell populations within post-sort fractions, providing details on the purity of the samples and how it was determined.* |
| Gating strategy | *Describe the gating strategy used for all relevant experiments, specifying the preliminary FSC/SSC gates of the starting cell population, indicating where boundaries between "positive" and "negative" staining cell populations are defined.* |

☐ Tick this box to confirm that a figure exemplifying the gating strategy is provided in the Supplementary Information.

# Magnetic resonance imaging

## Experimental design

| Design type | *Indicate task or resting state; event-related or block design.* |
|---|---|
| Design specifications | *Specify the number of blocks, trials or experimental units per session and/or subject, and specify the length of each trial or block (if trials are blocked) and interval between trials.* |
| Behavioral performance measures | *State number and/or type of variables recorded (e.g. correct button press, response time) and what statistics were used to establish that the subjects were performing the task as expected (e.g. mean, range, and/or standard deviation across subjects).* |

## Acquisition

| Imaging type(s) | *Specify: functional, structural, diffusion, perfusion.* |
|---|---|
| Field strength | *Specify in Tesla* |
| Sequence & imaging parameters | *Specify the pulse sequence type (gradient echo, spin echo, etc.), imaging type (EPI, spiral, etc.), field of view, matrix size, slice thickness, orientation and TE/TR/flip angle.* |
| Area of acquisition | *State whether a whole brain scan was used OR define the area of acquisition, describing how the region was determined.* |

Diffusion MRI     ☐ Used     ☐ Not used

## Preprocessing

| Preprocessing software | *Provide detail on software version and revision number and on specific parameters (model/functions, brain extraction, segmentation, smoothing kernel size, etc.).* |
|---|---|
| Normalization | *If data were normalized/standardized, describe the approach(es): specify linear or non-linear and define image types used for transformation OR indicate that data were not normalized and explain rationale for lack of normalization.* |
| Normalization template | *Describe the template used for normalization/transformation, specifying subject space or group standardized space (e.g. original Talairach, MNI305, ICBM152) OR indicate that the data were not normalized.* |
| Noise and artifact removal | *Describe your procedure(s) for artifact and structured noise removal, specifying motion parameters, tissue signals and physiological signals (heart rate, respiration).* |
| Volume censoring | *Define your software and/or method and criteria for volume censoring, and state the extent of such censoring.* |

## Statistical modeling & inference

| Model type and settings | *Specify type (mass univariate, multivariate, RSA, predictive, etc.) and describe essential details of the model at the first and second levels (e.g. fixed, random or mixed effects; drift or auto-correlation).* |
|---|---|
| Effect(s) tested | *Define precise effect in terms of the task or stimulus conditions instead of psychological concepts and indicate whether ANOVA or factorial designs were used.* |

Specify type of analysis:     ☐ Whole brain     ☐ ROI-based     ☐ Both

| Statistic type for inference (See Eklund et al. 2016) | *Specify voxel-wise or cluster-wise and report all relevant parameters for cluster-wise methods.* |
|---|---|
| Correction | *Describe the type of correction and how it is obtained for multiple comparisons (e.g. FWE, FDR, permutation or Monte Carlo).* |

## Models & analysis

| n/a | Involved in the study |
|-----|----------------------|
| ☐ ☐ | Functional and/or effective connectivity |
| ☐ ☐ | Graph analysis |
| ☐ ☐ | Multivariate modeling or predictive analysis |

**Functional and/or effective connectivity**

*Report the measures of dependence used and the model details (e.g. Pearson correlation, partial correlation, mutual information).*

**Graph analysis**

*Report the dependent variable and connectivity measure, specifying weighted graph or binarized graph, subject- or group-level, and the global and/or node summaries used (e.g. clustering coefficient, efficiency, etc.).*

**Multivariate modeling and predictive analysis**

*Specify independent variables, features extraction and dimension reduction, model, training and evaluation metrics.*

