## [Peer Review File · Nature]

Manuscript Title: Four ways blue foods can help achieve food system ambitions across nations

Reviewer Comments & Author Rebuttals

Reviewer Reports on the Initial Version:

Referees' comments:

Referee #1 (Remarks to the Author):

Crona et al. summarize and discuss, in an excellent contribution, the major implications of a large-scale project with focus on Blue Foods. The study uses existing datasets and expert knowledge to propose four policy objectives to enhance the global food systems using aquatic foods. The approach to derive these policy objectives heavily relies on expert knowledge and the general outcomes of the Blue Food Assessment. I am aware that some scholars are a bit uncomfortable with relying on expert knowledge; however, I am of the opinion that pieces such as this one by Crona are not only valid, but more importantly crucial for making significant progress on these complex topics. The original approach presented by the authors will undoubtedly be highly cited and of interest for researchers working on food systems, but also for decision- and policy-makers.

The authors made the data available through a web app that is accessible and easy to use. Similarly, the methodology is straightforward. I hope that the figures of the appendix can be made available through the same website as the x-axis of most of the figures in the appendix (figure 2 onwards) are literally impossible to read in the pdf. This is a minor technical point that can be easily solved. Given the straightforward methodological approach, the possibility for readers to "play" with the data, and the sound analysis, I find the conclusions robust and appropriate.

Further, the paper is very well written, although I make some suggestions below that could improve the readability, particularly in the sections that focus on the results. The other sections are outstanding and very clear to follow. The authors provided the right context and the selection of topics to discuss is timely, as are the references used. It is true that the analysis could be expanded to other topics, for example, other trade-offs could be analyzed; however, the authors clearly recognize the need for local settings for the relevance of the different trade-offs. Therefore, although additional analyses could be made, these analyses would target a specific jurisdiction, which would make the study less appealing for the broader readership of Nature.

Building upon the previous point, as a consequence of a global analysis such as the one carried out in this study, some nations could come to the conclusion that they would not benefit from policies, e.g. the policy objectives are mostly "less relevant" for Peru (only "Relevant" for Safeguard food system contributions) and Canada or Russia (only "Relevant" for Environmental footprints). Despite that, the policy objectives are crucial food for thought that decision- and policy-makers should consider no matter what, as every small contribution counts.

As you can derive from my previous comments, I enjoyed the work; however, there is always room for improvement:

- Section "Translating the blue food science into policy objectives": The policy objectives are the central piece of the work but they are not explained in the text, only in a table! This was a big surprise to me, and I checked several times to see if I missed some pages when downloading the pdf. I strongly believe that the four policy objectives should be briefly explain in the body of the document and not simply presented in a Table. Figures and Tables should support the text, but you cannot direct the reader to find such important content in a Table or Figure and that's all. This

could be an issue of space, but the four policy objectives are the core of the paper.

- I found Section 4 to be the weakest part of the paper. The section currently describes some of the metrics in Figure 2, but it does not "digest" the information for the reader. The last sentence of the section does not follow the flow from the previous one. There is a typo in line 194 as it should say 124 countries and not 118 (I think!). I think that this section should explain why some of the metrics of Figure 2 are described and others are not. For example, the 55% of Reducing deficiencies vs Safeguard food systems is not discussed, but lower percentages are described in detail. As I mentioned, the authors should justify why the specific interactions that they describe are relevant. In fact, I think that they have an excellent opportunity in this section to mirror the co-benefit section. The co-benefit section focuses strongly on environmental sustainability and climate resilience, which corresponds with the highest percentages in Figure 2.

- Figure 3: This figure is not cited in the document, which should be caught during the proofreading stage. In any case, it should be linked to the section "Domestic consumption vs export revenues". Further, the legend should be toned down. The use of "demonstrates" in line 335 is very strong. The figure does not demonstrate anything, perhaps it suggests. Also, the sentence "Scales indicate (im)balance..." is unclear to me.

Minor:

- The sections/subsections are a little bit weird. This will be solved at the editorial stage, but the first section with a number is section 3. Section 1 and 2 do not exist.

- I honestly do not like the style "Figure X shows..." at the beginning of a section. As I mentioned above, figures and tables should support the written section. When I read "Figure X shows..." I skip to the next section as the "Figure X shows..." can be extracted from the figure caption. It is easy, convenient, but not elegant.

- A good place to cut some words is in section 2, lines 145-154. I think that the example is not strictly needed and it could be further developed in the appendices. In the current draft, I think that this example is needed only because the policy objectives are not described in the text. As soon as the policy objectives are further developed, I don't think that the example will be needed.

In summary, I think that the paper by Crona is a great contribution. I greatly appreciate this opportunity to review the paper, and I think that after the results are polished to improve readability, it will be a valuable and interesting contribution for other readers.

Ramón Filgueira
Halifax, Canada

Referee #2 (Remarks to the Author):

This is a very interesting and relevant manuscript for giving context to the Blue Food movement. It does a very nice job discussing the potential tradeoffs that may occur when promoting the consumption of blue foods across the globe and leans into the idea of local context being important. This furthers the state of research of the field. One rather major issue I do see with the paper, is that it uses the term "sustainable" many times throughout, without providing a definition or context to the matter. Sustainability is all about time scales, with the idea that nothing, not even human life is sustainable indefinitely. For how often this term is utilized throughout the manuscript, an actual definition, context, and time scale needs to be provided.

A. The key result is presented in the figures, which is that there is the potential for the adoption of blue foods to help feed the world, and balance nutritional issues in a sustainable manner.

B. This is a novel approach, and well applied.

C. Data and methodology have no issues.

D. Discussion of uncertainties is included in the methods section.

E. Overall the conclusions are reasonable.

F. Minor comments:

Ln. 82-83, So reverse substitution has been observed. What makes substitution seem feasible in this context?

Ln. 91, "fed carp" seems a bit redundant here, as these are all "fed aquaculture systems"

Ln. 99, "a shift in feed composition", this seems to side step the significant issues due to the dependence on fish meal and oil from forage fish, particularly as the demand for aquafeeds continues to increase. "Lowering feed conversion ratios," admittedly this sounds like genetically engineering the fish, which is fine, but I would like to make sure that I am understanding this correctly, as it is somewhat ambiguous. Also, shifting feed compositions has a significant potential to change the nutritional benefits of the produced fish. So you need to be a bit careful here, if you are also making the case of nutritional benefits of eating blue foods, particularly Omega 3's and B12.

Ln 341, several times throughout the manuscript the importance of small scale producers is mentioned. However, the Covid-19 impacts doesn't seem to talk about this in any real fashion. Many of the small scale aquaponic producers in the Midwest provide their leafy greens and fish to schools and restaurants. Both of which were shuttered for parts of the pandemic and have resulted in these producers going out of business. There is the potential for this to be a significant loss to the industry.

Also, a working definition of what you mean by sustainability needs to be provided.

G. References are sufficient.

H. Clarity and context are acceptable.

Referee #3 (Remarks to the Author):

The manuscript by Prof Crona and colleagues presents a work integrating the findings from a worldwide initiative, the Bluefood assessment, focusing on the multiple roles of Bluefoods (sourced or cultivated in aquatic environments). The work investigates synergies and trade-offs associated with policy objectives, first examining and translating a set of recognized blue food functions into 4 policy objectives, and subsequently mapping the relevance of each objective to individual countries, based on public statistics. This allowed to obtain a map of national relevance for each of the four bluefood policy objectives, an overlap in relevance of the policy objectives, therefore discussed in terms of potential trade-offs associated with economic/nutritional benefits from the bluefoods. The tool developed for the analysis is provided as a web application, allowing to explore the effect of different cut-off values on the relevance of the policy objectives.

Given its focus, I feel that this work can be of immediate interest for a broad audience, beyond the specific fields of fisheries, aquaculture and nutrition. The focus on trade-offs associated to economic and nutritional benefits, the analytical framework and the online tool provided present a high potential for guiding future policies, and building scenarios for altering currently unsustainable practices of food production and consumption.

As recognized by the authors, the process of selection of the cut-off values was critical and the purpose of the exercise was to illustrate the difficulties connected with the analysis of national contexts (Ins 621-626).

With respect to this latter point, I feel that a couple of methodological aspects, which I tried to summarize here below, should be clarified and maybe present some margin for improvement. My main point of concern regards the background of the methodology used in the analysis and the sensitivity study.

- The Boolean approach used for selection was chosen in place of alternative approaches (e.g. based on fuzzy-logic), and the analysis considered the same weight for all the variables. I think that the choice of the methodology here must be justified, and framed in the context of the existing approaches available (methodological references should be provided);
- the one-at-a-time methodology was used in the sensitivity analysis. Is there any alternative methodology which could be used here? Sensitivity analysis is a well-established field, and I feel that also this part of the work should be better framed within the literature;
- an additional question is related to the differential impacts that specific climate hazards can have on freshwater and marine fisheries and aquaculture supply chains. The approach currently implemented pools together hazards impacting land and the ocean, and considering the overall country productivity. Is there any possibility for considering separately these impacts?

Specific comments:

Table S1. Column "variable". I suggest using consistently variable names between Table 1 and Table 2. (e.g. economic value/export revenue).

Table S2. In my view, this table should be placed before S1

Caption Data Figure S1. "This is because the two variables primarily differentiate...". This part was not fully clear to me.

Ln 10. Is there any overlapping/repetition with what presented at Ln 5?

Ln 96-100. "Blue food can... ..improved husbandry practices". Up to my understanding, this sentence mixes different concepts related to fisheries and aquaculture. I feel that it may be possible to state it more clearly.

Ln 105-106. "Smallscale operations..." I am not sure about the grammar here.

Ln 129. SFFA. The acronym was not introduced.

Ln 125-126. "We translate the blue food functions ...". More general comment about the structure. Is it functional to have such an important statement only at this point of the article?

Figure 1. If printed in grayscale "less relevant" and "missing data" are not distinguishable, maybe there is a possibility to change this.

Figure 2 caption. "Pol y" where y-axis policy is also relevant?

Lns 212 and 230. Citations are not ordered progressively. Was it done on purpose?

Figure 3. In the second row, I suggest inverting equity and environment, in order to be consistent with the legend. Also, the yellow colour (2nd row) is not mentioned in the legend. If printed in grayscale "less relevant" and "missing data" are not distinguishable.

Referee #4 (Remarks to the Author):

Review Blue Food policy objectives: an analysis of opportunities and trade-offs

The study presents a highly complex analysis of the global blue food resources aimed to provide a tool to policymakers to identify nationally relevant approaches to act to preserve and sustainably utilize these resources. The analysis presents an interesting and novel framework to understand the fundamental conflicting interests in the access to and use of blue food resources, such as the conflicts (trade-off) between small-scale vs industrial scale, nutrition-dense vs volume production, and climate optimized (single species) vs diversified resilient production. Also, the analysis highlights that any transformation of food systems is about the substitution of food sources, which is important to avoid simplified conclusions. Overall, this is an ambitious and potentially highly important analysis, presented by a highly qualified group of authors. In this view, I can

recommend the manuscript to be considered for publication, taking the concerns expressed below into consideration.

Provide examples of policies:

The analysis as presented serves as a guiding framework for policymakers to make the utilization of the rich global aquatic resources environmentally sustainable, and at the same time, support the nutrition and health of relevant populations. The foundation of the analysis is to highlight the diversity and complexity characterizing the unique value of the global blue food systems as a strength and at the same time, pointing out that these diverse systems are the roots of multiple conflicting interests (trade-offs). For the development of the policy framework, it appears as that the main purpose is to keep it simple, with only four overall objectives, of which the fourth – safeguarding contributions to nutrition, economics, and livelihoods – is cross-cutting the three others, adding the balancing of potential trade-offs, in particular with climate adaptations. Throughout the paper, the analysis highlights the specific importance of policies preserving the diversity and access of small-scale producers as a means of safeguarding resilience. While the intention of presenting a simple framework for policy objectives is an interesting first step to support policymakers to start the process of identifying national relevance and gaps of policies for sustainable utilization of blue food resources, the simplification into the four objectives also leaves many unanswered questions. Recognizing the aim is to provide the framework and not provide all the answers, it would anyway be more convincingly that the framework could actually be instrumental if the authors could provide a set of examples of types of policies, particular policies which could serve to balance the trade-offs of conflicting interests.

The policy objectives on nutrition and health:

The four policy categories as presented in the introduction are 'functions' to 'support blue food system transformation'. Of these four policy objectives, two are specifically targeted the potential of blue food to improve human health: 'Reducing deficiencies' and 'reducing cardiovascular disease risk'. While the presentation of the overall need for an analytical framework is argued with reference to the findings of the general blue food assessment, the justification of the underlying decisions and assumptions for these two policy objectives referring to nutrition and health is not so clear. In the introduction, lines 68-75, the nutrient objective is presented as blue food generally can reduce 'micronutrient deficiencies' without referring to any specific nutrients.

Only on page 7 the policy objective of 'reducing deficiencies' is specified to in this case to refer to inadequate intake of two specific nutrients, vitamin B12 and omega-3 (minor note: 'omega 3', should be written 'omega-3', the nomenclature is carbon no. 3 counted backward, minus 3). These are clearly relevant nutrients to be considered for deficiency, but there are others as well. Why these? The supplementary material provides the underlying assumptions and sensitivity analysis for the national estimation of deficiency (including the assumption to compensate for lack of agreed cut-offs) but the authors lack to argue for why these two nutrients were selected to represent 'micronutrient deficiencies'. Also the logic of merging B12 and omega-3 as 'micronutrient deficiencies' is just not clear. It could be a very different picture of national relevance if the nutrients were presented individually, and could lead to different policy implications. As likely built into the model, the Blue food sources differ in omega-3, such as marine sources are generally higher in omega-3 than freshwater sources, and for cultured species, the omega-3 highly depends on the feeding. So it is likely different blue food policies are needed to address B12 and omega-3 deficiency.

The criteria for inclusion and exclusion of countries are specified in the supplementary material, and other aspects are likely built into the general blue food assessment background material, but without the arguments for the specific nutrients, the value of the framework as a tool for policymaking is blurred. Is the framework is truly instrumental and is the outputs truly relevant for national policies? More elaborated arguments should be presented in the introduction. Or it leaves the reader with a good understanding of the relevance of guiding policies to incorporate the complexity and trade-offs in the blue food transitions ahead, but with an unclear picture of the real value of the approach as presented.

For the health objective: In the introduction, the health objective is presented as blue food is a 'healthy protein source' (alternative to red meat) which can contribute to reducing cardiovascular disease risk (line 76-85). The risk of cardiovascular diseases is relevant but as for the nutrition deficiencies, it is also selected among a range of dietary risk factors. It is understood that the framework can be applied to other risks following a set of inclusion and exclusion criteria, when data is available, but in any case - also the cases presented - the selection should be better argued. Also, the risk reduction of cardiovascular diseases related to aquatic foods is quite well documented to associate with LCPUFA (DHA, EPA) rather than to 'protein' as such, as indicated in line 76-78. It might be that the authors just used 'protein' as a synonym for food, but in this context, it confuses the message, especially because the fatty acids are also covered in the previous objective of reducing deficiencies. The protein reference is further disturbing by inserts mentioning plant proteins (line 77 and line 327-330). The framework builds on analyzing blue food substitution of meat, particularly red meat. If the authors also want to discuss blue food substituting plant-based foods, this should be fully incorporated and include also the nutritional trade-offs (like for omega-3 and B12). Or it should be left out, and not mentioned as it does not add to the context of the analysis.

Author Rebuttals to Initial Comments:

Response to reviewers - Nature manuscript 2021-02-03260B

To the Editor(s)

Thank you for allowing us to revise and resubmit the paper *Blue food policy objectives: an analysis of opportunities and trade-offs*. We have considered all the reviewers comments and have responded in detail in the submitted document. In addition to the reviewers comments we have also updated the functionality of the interactive data exploration site with the following.

- Added 2 tabs with PDFs – Supplementary table 2 (Boolean logic table) and supplementary figure 6 (data distributions). Added to create a more seamless experience using the app (i.e. so users do not have to refer back to paper). *NOTE: for now this is simply a pdf, but this will be revised and figures incorporated into the app when/if the paper is accepted*
- Updated the maps and various sizes/fonts etc for easier viewing
- Added an interactive table under the maps that represents the same information. Allows reader to easily get a list of countries with same policy outcome category, and improves information for small countries (hard to see on map)
- Added interactive text that highlights the contribution of BF to each country based on selections (policy and country) and the policy outcome. To aid in interpretation of results (tab Country breakdown)
- Added extra “dots” in the climate hazard visualization to highlight the variations in climate hazard score by sub-system and how these relate to the threshold – in part addresses the comments by reviewer 3.

We note that if the paper was to be accepted we would consider further work on this interactive tool prior to publication, and are happy to discuss any inputs from Nature with regard to this.

Sincerely,
Beatrice Crona on behalf of all authors

Comment	How it has been addressed
Reviewer 1 policy objectives are the core of the paper – need to be incorporated into main text (not just in table)	This is a valid and welcome point. In our effort to condense the text we erroneously omitted an elaboration of the policy objectives in the actual text. Given the constraints on words, we have attempted to find a solution that now introduce them under the subheading ‘translating blue food science...’, but where additional information and elaboration can be sought in Table 1. We hope this addresses the reviewer’s valid concern

Section 4: describes some of the metrics in Fig 2 but it does not “digest” the information for the reader. The last sentence of the section does not follow the flow from the previous one. There is a typo in line 194 as it should say 124 countries and not 118 (I think!). I think that this section should explain why some of the metrics of Figure 2 are described and others are not. For example, the 55% of Reducing deficiencies vs Safeguard food systems is not discussed, but lower percentages are described in detail. As I mentioned, the authors should justify why the specific interactions that they describe are relevant. In fact, I think that they have an excellent opportunity in this section to mirror the co-benefit section. The co-benefit section focuses strongly on environmental sustainability and climate resilience, which corresponds with the highest percentages in Figure 2.	Thank you for helping to make this section (hopefully) clearer. We now open the first § of this section with a sentence explaining that some countries exhibit conditions that make multiple policy objectives relevant – to set the stage. We then note that Fig 2 shows for which policy objectives such overlap exists and we make it clear that we will discuss overlaps that are particularly noteworthy (effectively all that are above 50%). We therefore include the 55% of Reducing deficiencies vs Safeguard food systems (noted by reviewer), which was simply an oversight on our part. It is true that this short section is somewhat of an analytical prelude to the next discussion on co-benefits, and we have tried to make this link clearer by now adding a final sentence to clarify that the subsequent section explores in more detail how potential co-benefits flagged by Figure 2 can be realized. We hope this gets at the point raised by the reviewer of ‘mirroring’ the co-benefit section, and perhaps better signposts for the reader. Thank you also for catching our typo – it should of course read 124.
Figure 3: This figure is not cited in the document, which should be caught during the proofreading stage. In any case, it should be linked to the section “Domestic consumption vs export revenues”. Further, the legend should be toned down. The use of “demonstrates” in line 335 is very strong. The figure does not demonstrate anything, perhaps it suggests. Also, the sentence “Scales indicate (im)balance...” is unclear to me.	Thank you for catching this omission. The figure is now cited in the introduction to the section on ‘Navigating tradeoffs...’ as a visualization of some example tradeoffs associated with pursuing economic or nutritional benefits of blue food (line 323). The figure has been substantially revised, also reflecting helpful comments by another reviewer. The legend has also been revised entirely, using more moderate language. It now reads: Figure 3. Example of hypothetical trade-offs associated with policies pursuing economic and/or nutritional benefits of blue food. The figure illustrates one set of trade-offs in policy outcomes that may result across the dimensions of environment, equity, economy and nutrition, depending on the degree of prioritisation of either increasing domestic blue food supplies for nutritional outcome, or maximizing economic revenue from blue food production from high value export markets. The degree of emphasis placed on either policy goal is represented by the blue bars. Likely outcomes for each

	dimensions are represented by coloured boxes and strength of outcome by +/- symbols; with positive outcomes depicted in green, and negative in pink. Sustainable commodification aligned with local preferences and demand represents an example of how a balance could be struck to optimize positive environmental, inclusive, economic and nutritional outcomes. Unknown impacts, or where policy objectives are judged to have a strong impact, are depicted in grey.
The sections/subsections are a little bit weird. This will be solved at the editorial stage, but the first section with a number is section 3. Section 1 and 2 do not exist.	Yes, we realize that the numbers of the section headings had gone missing in the submitted version. We apologize for this. We included numbered headings merely to facilitate the review process. These will be removed upon editorial guidance, should our work be accepted
I honestly do not like the style "Figure X shows..." at the beginning of a section. As I mentioned above, figures and tables should support the written section. When I read "Figure X shows..." I skip to the next section as the "Figure X shows..." can be extracted from the figure caption. It is easy, convenient, but not elegant.	We acknowledge the reviewer's request and have attempted to rework the text to avoid this style of writing.
A good place to cut some words is in section 2, lines 145-154. I think that the example is not strictly needed and it could be further developed in the appendices. In the current draft, I think that this example is needed only because the policy objectives are not described in the text. As soon as the policy objectives are further developed, I don't think that the example will be needed.	Thank you for identifying places to reduce words without hopefully reducing readability. This is much appreciated. As noted above, we have moved a revised version of the example up to exemplify how we translate blue 'food functions' into policy objectives (now under 'Translating blue food...'). We feel the example serves a purpose in exemplifying the process of translation and use of nationally available statistics as proxy variables. Helping readers understand this is key to the credibility of the remainder of the paper. This nonetheless reduced words, and helped us reduce redundancies in the text, so thank the reviewer for that.
Reviewer 2	
One rather major issue I do see with the paper, is that it uses the term "sustainable" many times throughout, without providing a definition or context to the matter. Sustainability is all about time scales, with the idea that	We acknowledge and agree that the term 'sustainability' is often used without clear reference or definitions, and we had not done enough to clarify our use of the term.

nothing, not even human life is sustainable indefinitely. For how often this term is utilized throughout the manuscript, an actual definition, context, and time scale needs to be provided	To remedy this we have elaborated on the meaning of the term sustainability in the context of this paper on page 6. The reason for inserting it here is that an elaboration of the term where it is first used (1st § of intro) would significantly cut up the flow of the text. Instead we have specified that we are referring to ‘environmentally sustainable’ on line 26 to make our use of the term more precise. This specification of the environmental aspect of sustainability has been done throughout the paper, and wherever possible we have also changed the wording to specify our meaning by using phrases such as ‘low environmental impact’, ‘less environmentally harmful’ or ‘less environmentally impactful’. We have also specified environmental sustainability in the abstract.
Ln. 82-83, So reverse substitution has been observed. What makes substitution seem feasible in this context?	Thank you for this important comment. While recent work shows that seafood, up till now, does not seem to have significantly displaced red meat (e.g. see York 2021), we’re in a new era of rapid societal transformation fueled by a widespread societal concern for staying within certain planetary boundaries and meeting the Paris agreement. Changing diets is one important step towards achieving this and reducing the consumption of red meat is a documented necessity to drastically halt GHG emissions from the food system. We therefore believe that basing discussions of the role blue foods could have in the future food system solely on backcasting studies and past dietary trends is not sufficient. Earlier work has demonstrated that there is a gradient of preferred meat substitutes where fish/seafood is the most popular substitute for terrestrial meats (Schösler et al. 2012). Moreover, the general trend over the last 60 years has been a substantial increase in poultry consumption while beef consumption has stagnated (Naylor et al. 2021), suggesting that white meats can replace red meat. Given that blue foods (and other animal products, including poultry, egg and cheese) are generally consumed on a regular basis and part of the cuisine in many high meat-eating countries, they have the potential to constitute a stepping-stone for de-routinizing red meat

	consumption and eventually enable a shift to more plant-based diets. We have tried to clarify this on line 100. Naylor, Rosamond L., Avinash Kishore, U. Rashid Sumaila, Ibrahim Issifu, Blaire P. Hunter, Ben Belton, Simon R. Bush et al. (2021) Blue food demand across geographic and temporal scales. Nature communications 12, no. 1 1-14. Schösler, H., De Boer, J., & Boersema, J. J. (2012). Can we cut out the meat of the dish? Constructing consumer-oriented pathways towards meat substitution. Appetite, 58(1), 39-47. York, Richard (2021) Poultry and fish and aquatic invertebrates have not displaced other meat sources. Nature Sustainability 4, no. 9: 766-768.
Ln. 91, “fed carp” seems a bit redundant here, as these are all “fed aquaculture systems” Ln. 99, “a shift in feed composition”, this seems to side step the significant issues due to the dependence on fish meal and oil from forage fish, particularly as the demand for aquafeeds continues to increase. “Lowering feed conversion ratios,” admittedly this sounds like genetically engineering the fish, which is fine, but I would like to make sure that I am understanding this correctly, as it is somewhat ambiguous. Also, shifting feed compositions has a significant potential to change the nutritional benefits of the produced fish. So you need to be a bit careful here, if you are also making the case of nutritional benefits of eating blue foods, particularly Omega 3’s and B12.	Thank you, yes a superfluous word indeed. ‘fed’ has been removed Lowering of food conversion ratios (FCR) relates primarily to genetic improvements accomplished through breeding programs. This will improve growth and uptake of feeds and a reduced environmental footprint. We have rewritten the final sentences of this §, inserting the example of breeding programs to clarify this point, and also exemplifying how the shift in feed composition that is referred to is in fact a shift away from aquatic protein, to alternative feed sources, some of which are receiving a lot of R&D at the moment (like insect meal). Such changes in feed composition can result in a significantly reduced environmental footprint. For example, if soy originating from land where rainforests have been cleared is replaced by deforestation free soy, greenhouse gas emissions of aquaculture production can be reduced by up to 54% (Gephart et al. 2021. Environmental performance of blue foods. Nature) However, the relationship between fish meal and oil in aquafeeds and how this relates to sustainability is complex and does not lend itself to easy generalizations. For example the use of fish resources in feeds that originate from energy efficient fisheries, i.e. fisheries targeting small pelagic fish, generally results in a lower GHG footprint than most terrestrial alternatives.

	Using fish resources and seafood processing wastes from sustainably managed fisheries for aquafeed ingredients thus has the capacity to reduce the environmental footprint. However, even if a fishery is well managed and fished at sustainable levels it still implies transforming food resources that could be fit for human consumption to animal feeds. We recognize this is a much larger topic of discussion, which we are not able to delve into in this section (or paper). We therefore focus on clarifying (using examples) how our suggestions could reduce environmental impact, thereby hoping to reduce the ambiguity noted by the reviewer On shifting feed compositions: Thank you for noting this important trade-off. We do acknowledge this potential issue in a few instances in the paper. For example, it is flagged in the example column of Table 1 (1st row) "Nutrition-sensitive trade, processing, and distribution policies are also important, to avoid a scenario where increased aquaculture production delivers foods that are not nutrient rich, affordable or accessible to those who need it" and "Trade-offs between the environmental impacts of feed production and the nutritional quality of the fish produced also need to be assessed.", as well as in col 4, row 3 of the same table We also explicitly discuss this under section 5 (Sustainability vs. nutritional content of aquaculture), where we note that by incorporating plant-based ingredients and recycled animal processing wastes in feeds to reduce environmental footprints, there is a risk of compromising the nutritional value of the fish produced. Thus we hope that the potential for a trade-off has been sufficiently flagged for readers to realize this warrants attention when working to reduce environmental impact of blue food production.
Ln 341, several times throughout the manuscript the importance of small scale producers is mentioned. However, the Covid-19 impacts doesn't seem to talk about this in any real fashion. Many of the small scale	Thank you for this thoughtful insight. Certainly, as we begin to better understand the impacts of the pandemic a wide range of cases where small-scale actors were negatively affected are being documented (e.g. Campbell et al 2020, Belton et al 2021). We also see studies demonstrating the vital

aquaponic producers in the Midwest provide their leafy greens and fish to schools and restaurants. Both of which were shuttered for parts of the pandemic and have resulted in these producers going out of business. There is the potential for this to be a significant loss to the industry.	positive roles the small-scale sector played in some contexts, such as filling nutritional gaps caused by the interruption of globalized value chains (e.g. Bennett et al 2020). Our mention of covid-19 was merely intended to highlight the potential impacts of such large-scale shocks, rather than as a commentary on the pandemic itself, as this is not a focus of the manuscript. However, the impacts on the small-scale sector flagged by the reviewer is an excellent example of the types of radical changes we are referring to, both in positive and negative ways, so we have added a sentence to bring in this valuable insight in more clearly on line 393-395. Bennett, N.J., Finkbeiner, E.M., Ban, N.C., Belhabib, D., Jupiter, S.D., Kittinger, J.N., Mangubhai, S., Scholtens, J., Gill, D., Christie, P., 2020. The COVID-19 Pandemic, Small-Scale Fisheries and Coastal Fishing Communities. Coastal Management 1–11. https://doi.org/10.1080/08920753.2020.1766937. Belton B, Rosen L, Middleton L, Gazali S, Mamun AA, Shieh J, Noronha, HS, Dhar G, Ilyas M, Price C, Nasr-Allah AM, Elaira I, Ballarsingh BK, Padiyar A, Rajendran S, Mohan AB, Babu R, Akester MJ, Phyoo EE, Khin MS, Olaniyi A, Siriwardena SN, Bostock J, Little DC, Phillips MJ and Thilsted SH. 2021. COVID-19 impacts and adaptations in Asia and Africa’s aquatic food value chains. Penang, Malaysia: CGIAR Research Program on Fish Agri-Food Systems. Working Paper: FISH-2021-02. Campbell, S.J., Jakub, R., Valdivia, A., Setiawan, H., Setiawan, A., Cox, C., Kiyo, A., Darman, Djafar, L.F., Rosa, E. de la, Suherfian, W., Yuliani, A., Kushardanto, H., Muawanah, U., Rukma, A., Alimi, T., Box, S., 2020. Immediate impact of COVID-19 across tropical small-scale fishing communities. Ocean & Coastal Management 105485. https://doi.org/10.1016/j.ocecoaman.2020.105485
Reviewer 3	
- The Boolean approach used for selection was chosen in place of alternative approaches (e.g. based on fuzzy-logic), and the analysis considered the same weight for all the variables. I think that the choice of the methodology here must be justified, and framed in the context of the existing	We appreciate the feedback from the reviewer regarding this point. By better clarifying the origins of our methods, specifying our justification for certain choices and connecting it to relevant literature, we hope it both strengthens the legitimacy of our methodological choice as well as articulates the methodological contribution. We are grateful to the reviewer for pushing this point, and improving the paper. Below we outline in detail the origins of our methodological approach, and explain our choices and reasoning. The most

approaches available (methodological references should be provided);

pertinent aspects have been brought into the methods section (which is now significantly revised), but given word constraints the methodological background is instead elaborated briefly in the supplementary materials. Our method can be described as an attempt at assessing policy relevance through backward QCA outcome matching. The QCA method and approach is often used in policy analysis, for example in analyses of policy formulation, implementation, and evaluation (Thomann, 2020), and was therefore close at hand for analysis on policy relevance. It is a theory driven analytical method for assessing the causal contribution of different conditions to outcomes through the concept of sets and their relations (Rihoux and Ragin 2008). A crisp-set QCA analysis traditionally follows three steps. The first step is building a dichotomous data table of conditions. In QCA, conditions are variables that could logically explain an outcome (Rihoux & Meur, 2009), and where the outcome is the object of study. In traditional QCA analysis, the outcome is a known variable in the data, which the conditions are used to explain, an example is the existence of a strong welfare state (outcome) with an analysis of conditions such as a strong left party and strong unions (Grofman & Schneider, 2009). If the conditions are continuous variables, they are dichotomised (to 0 and 1) based on a threshold. "Best practice" for dichotomising thresholds are to 1) be transparent with justifications for thresholds, 2) if possible determine threshold based on substantive or theoretical grounds, or if this is not possible 3) base thresholds on distributions along a continuum (Rihoux & Meur, 2009). In the revised methods we have articulated that our analysis is inspired by QCA, and that the first step of our analysis (which is concurrent with the first step in conventional QCA) adheres to the best practices outlined by leading scholars in the QCA field.

The second step of conventional QCA is matching the data table with the outcome data, resulting in what is referred to as a 'truth table'. The truth table shows every configuration of the data variables that lead to the outcome. The third step of a QCA is converting the truth table information into solution formulas. Solution formulas are based on Boolean logic and explain the matching of the conditions with the outcomes through Boolean AND, OR and NOT statements. The solution formulas highlight a key feature

and strength of QCA, namely the capacity to distinguish between necessary and sufficient conditions, something that cannot be determined through quantitative methods, such as regression models. Other strengths of the QCA method include the emphasis on equifinality, multifinality and asymmetric causality (Grofman & Schneider, 2009). Equifinality is the fact that different combinations of conditions can lead to the same outcome. Multifinality is that the same conditions can play different roles in different contexts or situations. Finally, asymmetric causality is the fact that the conditions that lead to the occurrence of an outcome, do not have to be the same as those that lead to the non-occurrence of an outcome (Grofman & Schneider, 2009). The strengths of the QCA methodology, as well as its common usage in policy analysis, formed the basis for our inspiration to draw on this method to create the novel methodology in this article. Our adjusted methodology, retains the ability to convey a form of equifinality, multifinality and asymmetric causality. To exemplify: the fact that the same condition, such as blue food availability, represents different things in the different policy objectives (such as providing nutrition or representing lower emission production) highlights multifinality. The different conditions that can help safeguard food system contributions under climate change (employment, export revenue or high consumption) highlights equifinality. Finally, that the solution formula for the outcome category “less relevant” is not the reverse of formulas for “relevant” and “highly relevant”, indicating asymmetric causality.

Our methodology follows three steps that each correspond to the different steps in the QCA methodology. However, our approach differs from a conventional QCA in that instead of starting with an empirically observed outcome variable and matching it to causal variables to arrive at a set of Boolean solution formulas, our analysis departs from a set of pre-defined Boolean solution formulas, matches these to the data configurations (step 1) and *generates* an outcome variable. This outcome variable is the classification of countries into four categories (‘highly relevant’, ‘relevant’, ‘less relevant, and ‘missing data’) that capture the “degree of policy relevance” (Table S2). The second step is the development of a set of Boolean logic solution formulas to be used in classifying countries based on the degree to which they exhibit the conditions specified in step one. These solution formulas are based

on expert judgement and logic (Table S2). The combination of AND and OR statements highlight the distinction between necessary and sufficient conditions, akin to a conventional QCA noted above (Rihoux and Ragin 2008). For data files used as input for the Boolean analysis see:

<https://doi.org/10.7910/DVN/ILA0XI>. R code used for Boolean analysis is available at: https://github.com/emmywas/BFA_Policy_analysis

In the third step, which we refer to as “backward outcome matching”, we match the set configurations in the data table (step 1) with the Boolean logic solution formulas designed in step 2 to assign each case (country) to the outcome variable ‘Degree of policy relevance’ (Table S2). This outcome variable (for each policy objective) forms the results of our QCA-inspired analysis.

On fuzzy versus crisp sets:

We use a crisp-set methodology. Fuzzy sets do allow for more information to be captured by allowing each case to have membership in more than one group (i.e. not binary 0 and 1) (Rohlfing 2020), however it relies on several thresholds being set, and assigning cases to a degree of membership for each group. Given the limitation on theoretically or empirically founded thresholds noted earlier, we opted for crisp sets to allow for a higher degree of transparency in the methodology. It is also possible, in the web-based tool, to change the threshold values and see the change in outcome. We have justified our choice of crisp sets in the revised methods text.

In the Supplementary Information we have added a new figure (Figure S1) which is a description of our methodological approach and comparison to conventional qualitative comparative analysis (QCA)

Grofman, B., & Schneider, C. Q. (2009). An introduction to crisp set QCA, with a comparison to binary logistic regression. *Political Research Quarterly*, 62(4), 662–672. <https://doi.org/10.1177/1065912909338464>

Rihoux, B., & Ragin, C. C. (2008). *Configurational comparative methods: Qualitative comparative analysis (QCA) and related techniques*. Sage Publications.

	Rihoux, B., & Meur, G. de. (2009). Crisp-Set Qualitative Comparative Analysis (csQCA). In Configurational Comparative Methods: Qualitative Comparative Analysis (QCA) and Related Techniques (pp. 33–68). SAGE Publications, Inc. https://doi.org/10.4135/9781452226569.n3 Rohlfing, I. (2020). The choice between crisp and fuzzy sets in qualitative comparative analysis and the ambiguous consequences for finding consistent set relations. Field Methods, 32(1), 75–88. Thomann, E. (2020). Qualitative Comparative Analysis for comparative policy analysis. In Handbook of Research Methods and Applications in Comparative Policy Analysis (pp. 254–276). Edward Elgar Publishing. https://doi.org/10.4337/9781788111195.00023
- the one-at-a-time methodology was used in the sensitivity analysis. Is there any alternative methodology which could be used here? Sensitivity analysis is a well-established field, and I feel that also this part of the work should be better framed within the literature;	We thank the reviewer for this critical question which triggered us to do a thorough review of the various options for a more formal sensitivity analysis. Given that we use the QCA inspired method to classify countries into categories, our sensitivity analysis needs to assess how consistently nations get classified the same way as when our set thresholds are applied. To achieve this we looked into the ways of employing a methodology that accounts for variation in all of the variables at once (for each policy). We settled on a modified simple random sampling sensitivity analysis (Hamby 1994), which consists of drawing a random number (sampled from a uniform distribution within the range of the data variable) for each threshold and running the classification model. The classification model was run 50 000 times for each policy to investigate the distribution in classification outcomes at each threshold level and compare them to our set thresholds. Since all thresholds are drawn randomly, this sensitivity analysis tests the influence of multiple changes to our threshold decisions simultaneously. The results of this analysis is attached for the perusal of the reviewer (see: Additional analysis provided to support our argument relating to sensitivity analysis). We discussed which sampling distributions and data ranges to use (and tested different options), but once the plots were produced it became clear that the patterns shown are very complex and difficult to interpret.

First of all, since the analysis builds on a high number of model runs with thresholds sampled randomly from the whole data range (as for e.g. blue food availability) or for the whole range of possible but not realistic values for each variable (such as for summary exposure values) it creates a lot of noise. For blue foods the observed data range from 0 to 4000, but the bulk of the observed data are at values between 0-20kg/cap/yr. Capping the data range or transforming the data is of course doable, but would then introduce additional theoretical or expert informed decisions on why it was capped at a particular value or why a certain transformation was chosen. Second, the patterns of the plots are not entirely intuitive because of the complex interaction that occurs, even though the Boolean logic rules are relatively simple. As an example one can look at the plots for sensitivity analysis for policy 1 relating to nutrient deficiencies (Supporting Figure 1 below). Here we can see that the white band in the vitamin B12 plot is an effect of the fact that at a particular value for omega-3 (at around 8%) approximately 30 countries suddenly fall above or below the value of x (where x is the set threshold value). however, this is not intuitive, and to understand it requires a significant amount of understanding of both the Boolean rules, the underlying data distribution and the capacity to interpret the plots.

Together, these realizations led us to the overall assessment that pursuing this line of sensitivity analysis would lead to less transparent and less easily understood results. Instead we concluded that the fact that the sensitivity of the threshold is most straightforwardly explained and understood by showing the chosen thresholds in relations to the distribution of the countries' data, it makes the most sense to simply provide these (see Figure S6). Having gone around one entire circuit we therefore came back to the more simple one-at-a-time analysis. We hope this lengthy (sorry) explanation and provision of our different analytical approaches makes it clear to the reviewer that we have taken the critique to heart and have attempted to thoroughly address it, but have landed in retaining a more simple transparent approach to show sensitivity.

For the purpose of the paper, we now therefore include (in the SI) histograms of the underlying data for each variable, and superimpose our set threshold.

	We also include a set of new revised plots (for each policy objective), representing a one-at-a-time analysis but now showing all three categories of relevance at once (see Figures S2-S5) – in contrast to our initial one-at-a-time analysis where ‘highly relevant’ and ‘relevant’ had been combined. Hamby, D. M. (1994). A review of techniques for parameter sensitivity analysis of environmental models. Environmental Monitoring and Assessment, 32, 135–154.
- an additional question is related to the differential impacts that specific climate hazards can have on freshwater and marine fisheries and aquaculture supply chains. The approach currently implemented pools together hazards impacting land and the ocean, and considering the overall country productivity. Is there any possibility for considering separately these impacts?	Thank you for this suggestion. It is true that within the diversity of aquatic food systems, different climate hazards differentially impact different production systems, and estimates of those differential impacts exist (Tighehaar et al., 2021). However, for purposes of this analysis, we decided to base our assessment on climate hazard across aquatic food systems, weighted by the production portfolio in each country. The main reason for this is that many of the national-level statistics on nutrition, employment, and revenue are not split out in marine versus freshwater and/or in fisheries versus aquaculture. This makes it very difficult to relate climate hazards to dependency on a system-by-system level. We therefore regard this analysis of policy relevance as an entry point into more granular decision-making around investments in climate resilient aquatic food systems. To aid in this, we have added graphical representation of the climate hazard scores for different sub-components of aquatic food systems to the interactive web tool that appear when the policy on ‘safeguarding food system contributions’ is selected (see tab ‘Country breakdown’). We also specify that analysis is based on present-day production portfolios in Table S1
Table S1. Column “variable”. I suggest using consistently variable names between Table 1 and Table 2. (e.g. economic value/export revenue).	Thank you for noting the inconsistencies in the terminology between these tables. We have now gone through Table S1 and aligned variable names (col ‘variable’) with the terms used in Table 1 and Table S2
Table S2. In my view, this table should be placed before S1	We have numbered the supplementary tables in the order they appear in the main text. With the revisions Table S1 is the first of the supplementary tables to be references and we have therefore chosen to retain it as Table S1.
Ln 96-100. “Blue food can... ..improved husbandry practices”. Up to my understanding, this sentence mixes	We agree there was scope for confusion given our wording. In order to more clearly distinguish between capture fisheries and aquaculture, we have now

different concepts related to fisheries and aquaculture. I feel that it may be possible to state it more clearly.	separated this sentence in two: the first relating to fishers and the second to aquaculture. We hope that this distinction makes it clearer.
Ln 105-106. "Smallscale operations..." I am not sure about the grammar here.	The sentence reads: "Small-scale operations provide two-thirds of aquatic foods destined for human consumption globally" We are unsure as to what the grammatical concern is, but happy to edit if we can have more clarification.
Ln 129. SFFA. The acronym was not introduced.	Thank you for spotting this inconsistency. SFFA was doubly wrong, as it was a misspelling of SSFA (small-scale fisheries and aquaculture, introduced later in the paper), but this term had not been introduced/spelled out before. We have therefore changed the wording.
Ln 125-126. "We translate the blue food functions ..." . More general comment about the structure. Is it functional to have such an important statement only at this point of the article?	In the last § of the introduction, we attempt to make it clear what the reader can expect, by stating: "This paper integrates the findings of an initiative to assess the multiple roles blue foods play in food systems around the world (the Blue Food Assessment: https://www.bluefood.earth/) and translates them into four policy objectives that could help realize positive health, environment and equity contributions of aquatic foods in food systems worldwide (our underline)." With hard constraints on words used we felt this flags the central contribution of reviewing literature and translating it into plausible policy objectives relating to blue foods. We then merely reiterate that the review of the role blue food can feasible play in the broader transformation of food systems (section 2), forms the basis for articulating a set of blue food policy objectives. The comment by R2 touches on our neglect to spell out the four policy objectives (also noted by R1), that are core to the paper. We hope that by addressing R1s comment (see above) we have also have addressed the concern of R2
Figure 1. If printed in grayscale "less relevant" and "missing data" are not distinguishable, maybe there is a	Yes, we acknowledge that this may be an issue but look forward to discuss with editors what is the best approach, should the article be accepted for publication.

possibility to change this.	
Figure 2 caption. “Pol y” where y-axis policy is also relevant?	Thank you for catching this mistake. The legend now reads: Parentheses in top row represent total number of countries where each policy is relevant. Each cell shows the number of countries (in parentheses) for which both column and row-heading policies are relevant, as a proportion of countries where the column-heading policy is relevant. Relevance in this figure indicates countries categorised as ‘highly relevant’ or ‘relevant’ for a given policy.
Lnns 212 and 230. Citations are not ordered progressively. Was it done on purpose?	This must be due to some bug in the referencing program. Will be amended
Figure 3. In the second row, I suggest inverting equity and environment, in order to be consistent with the legend. Also, the yellow colour (2nd row) is not mentioned in the legend. If printed in grayscale “less relevant” and “missing data” are not distinguishable.	Thank you for helping to make this figure clearer. In fact, based on this and constructive inputs from other reviewers this figure has been completely revised, thus dealing with the inconsistency noted by the reviewer.
Reviewer 4 Recognizing the aim is to provide a framework and not provide all the answers, it would be more convincing that the framework could actually be instrumental if the authors could provide a set of examples of types of policies, particular policies which could serve to balance the trade-offs of conflicting interests.	Thank you for this though provoking suggestion. As an author team, this is something we deliberated on extensively before submitting our paper. In fact, we initially embarked on a collection of policies from around the world to provide examples of policy objectives that try to address some of the noted trade-offs or promote synergies. However, we realized that there were (in our opinion) two significant risks with dwelling too much on these examples or providing too many details. These risks relate to the representativity of the policy examples discussed, and to the risk of directing too much attention to certain policy options and inadvertently appearing to recommend these as predominant or desirable options. More specifically, we believed there is a real risk of individual countries being singled out and erroneously compared to other countries by readers, especially since the focus and scope of the paper did not allow for a full-scale systematic review of national policies, across all the domains we touch on in the paper.

In other words, by providing a more thorough treatment of a non-systematically sampled set of policies, this particular set could risk being misconstrued as either direct recommendations or as a presentation of some predominant options – when in fact some policy options for addressing synergies or tradeoffs may have inadvertently been omitted. We therefore judged it to be more prudent to leave the suggestions at the higher level of policy goals. Sections 5+6 (on synergies and tradeoffs) provide some examples of broad policy options under each subsection, backed with some references. These were intended to provide exactly that – some broad examples of potential relevance across many different cultural or economic context – without providing too much of a specific country context and thus erroneously putting the focus on that nation or context. Prompted by Reviewer 4’s comments we went back over the text and found a certain imbalance, where examples for some trade-offs were lacking. To ensure that every subsection does in fact have examples flagged, we have now added some (demonstrated in lines 260-265, 274-276, 284-290, 304-315, 331-337, 350-356, 368-376). These have been vetted by co-authors to reflect diversity of policy approaches, but will naturally not represent a systematic selection.

With regards to the overarching aim of the paper, we are grateful that the reviewer recognizes our ambition to not provide all the answers, but rather provide a first step in bringing together the multiple dimensions of blue foods and present a framework that allows nations to get a broad assessment of relevance across the four blue food policy objectives we focus on. It is our hope that in providing this first broad assessment we can trigger policy makers, policy analysts and scholars alike, to delve deeper and make more thorough and granular analysis of the role blue foods could play for addressing multiple goals in specific national or subnational contexts.

We regret that we cannot provide a more refined decision support tool – which we believe is what the reviewer is asking for, but unfortunately a major constraint lies in the lack of publically available data at both national and

the presentation of the overall need for an analytical framework is argued with reference to the findings of the general blue food assessment, the justification of the underlying decisions and assumptions for these two policy objectives referring to nutrition and health is not so clear. In the introduction, lines 68-75, the nutrient objective is presented as blue food generally can reduce ‘micronutrient deficiencies’ without referring to any specific nutrients. Only on page 7 the policy objective of ‘reducing deficiencies’ is specified to in this case to refer to inadequate intake of two specific nutrients, vitamin B12 and omega-3 (minor note: ‘omega 3’, should be written ‘omega-3’, the nomenclature is carbon no. 3 counted backward, minus 3). These are clearly relevant nutrients to be considered for deficiency, but there are others as well. Why these? authors do not argue for why these two nutrients were selected to represent ‘micronutrient deficiencies’.	subnational levels. These kinds of data may be available in specific countries, or to public policy actors in certain regions and could constitute an excellent continuation of our work, by providing the means to conduct finer grained case-based analysis. However, currently the lack of data means a global analysis, such as the one pursued, here would not be feasible.
We agree with the reviewer that we have been too general in our use of the term micronutrient deficiencies in a number of places. We have worked to address this, and specifically in the section identified by the reviewer, we have specified that the nutrients we are particularly concerned with here are B12 and omega-3. We have added the justification for this. This justification is based on the fact that while blue foods clearly contribute a range of nutritionally important micronutrients, as shown by e.g. Golden et al (2021), their models (based on the most comprehensive dataset we are aware of) show that by 2030, blue foods may contribute a global average of 27.8% of vitamin B12, and 100% of EPA and DHA fatty acids. DHA+EPA content is usually lower in freshwater fish, but not zero (see e.g. Shepon et al 2022). For other nutrients, such as protein, iron, zinc and vitamin A, the total contribution by blue foods is lower, and deficiencies can arguably also be addressed through consumption of a range of other foods. Furthermore, these two (B12 and omega-3) do play an important role in foetal and child growth/development and risk of deficiencies (such as for B12) are particularly elevated for children and elderly. We hope this also helps to early on in the paper, indicate more clearly why our subsequent analysis is restricted to these two micronutrient deficiencies. Shepon, A., Makov, T., Hamilton, H. A., Müller, D. B., Gephart, J. A., Henriksson, P. J., ... & Golden, C. D. (2022). Sustainable optimization of global aquatic omega-3 supply chain could substantially narrow the nutrient gap. Resources, Conservation and Recycling, 181, 106260.	We have also added the justification to clarify our choice in Table 1 Finally, the omega-3 nomenclature has been corrected throughout the paper and extended data. Thank you for noting this mistake.

Also the logic of merging B12 and omega-3 as 'micronutrient deficiencies' is just not clear. It could be a very different picture of national relevance if the nutrients were presented individually, and could lead to different policy implications. As likely built into the model, the Blue food sources differ in omega-3, such as marine sources are generally higher in omega-3 than freshwater sources, and for cultured species, the omega-3 highly depends on the feeding. So it is likely different blue food policies are needed to address B12 and omega-3 deficiency.

This is a good and insightful point. The reviewer is right that by effectively lumping the deficiencies of vitamin B12 and omega-3 in the Boolean logic used to assess country relevance all we can say is that that 'relevant/highly relevant' countries are countries that have deficiencies above a certain level, for either or both of these micronutrients.

In making the decision to pursue this analytical path a number of considerations were made as we deliberated over the pros and cons. We have tried to outline them below, to explain and justify our choice.

It is true that we could have split this analysis into two identical types of analyses for B12 and omega-3, thus ending up with two sub-analyses for this policy objective. As the reviewer notes, this may have given a somewhat different categorization of national relevance, but most likely the main difference would be in the articulation of how to address the deficiency. I.e. deficiencies in omega-3 would mean a country would need to seek supplies of marine blue food, while deficiencies in vitamin B12 is not constrained to marine species.

Since vitamin B12 and PUFAs have a strong overlap in the health contribution they provide – notably they are recognized as particularly important for physiological and cognitive development – we argue that for the purpose of this paper they can be said to have a very similar 'blue food function'. As noted by the reviewer above (and elaborated by us above) the purpose of the paper is not to provide a detailed decision support tool for individual countries. Instead we aim to provide a first broad framework to begin to delineate potential blue food policy relevance to countries based on the functions blue foods can play in a particular context. A country which is flagged as potentially relevant for this particular policy in our analysis would need to check if they are low on omega-3 or vitamin B12 or both. This can be easily done in the interactive tool we provide. Based on this, they would then have to consider if their needs could be met by sourcing any aquatic foods, or specifically marine sourced. In this sense the analysis will not provide a detailed policy response to individual countries, but serves as a first indication.

	Building a more sophisticated and detailed model for analysis is hampered by the low resolution of much publically available data reported by countries, and by the diversity in contextual nuances that must be considered in national policy development. For example, deficiencies are likely to differ significantly across population segments, perhaps even subnational regions, and be associated with other factors like socio-economic status, etc. Many of these factors (and their subnational variation) would be important to consider when developing actual policy responses, and the model framework we provide here could never be fine-grained enough to serve this purpose. In light of these different consideration, and pros and cons, we feel strongly that a simpler approach with as much analytical transparency as possible is to be preferred. We hope it is clear why we chose not to treat the two micronutrients separately, as we wanted to reduce the number of layers of analysis the reader would have to penetrate. In other words, reduce the complexity of the analysis and also the message and ease of interpretation. One could argue that a simple but misleading message does not serve anyone, however we do not feel the message is misleading for two reasons. One is that the noted health contribution is similar, and the second is that any country flagged as relevant for a policy can easily check on their values for underlying variables in the interactive tool.
Introduction: needs more elaborated arguments for why the specific nutrients. Or it leaves the reader with a good understanding of the relevance of guiding policies to incorporate the complexity and trade-offs in the blue food transitions ahead, but with an unclear picture of the real value of the approach as presented. Same as for deficiencies: need to argue for why you chose to focus on cardiovascular diseases which is just one among a range of dietary risk factors.	On lines 74-81 we have now specified the micronutrients in focus, as flagged by the reviewer, and justified this focus, with the hope that this clarifies the focus on these particular nutrients, but also the specific nutritional value addressed.
	Under “Healthy alternatives to terrestrial animal protein” we have now changed the wording to note that (as the reviewer rightly points out) that blue foods can be leveraged to reduce dietary risk factors related to several NCDs, including cardiovascular disease, diabetes and various cancers (Wolk 2017, Richi et al 2015, Miller et al 2022). We have revised and updated the

	referencing of this section, and clarified that as cardiovascular disease is among the most commonly cited negative health effects of red meat consumption (Wolk 2017, Richi et al 2015) we use it as an example of how countries can assess the relevance of blue food policies depending on their specific conditions. Miller, V., Micha, R., Choi, E., Karageorgou, D., Webb, P., & Mozaffarian, D. (2022). Evaluation of the Quality of Evidence of the Association of Foods and Nutrients With Cardiovascular Disease and Diabetes: A Systematic Review. JAMA network open, 5(2), e2146705-e2146705. Richi, E. B., Baumer, B., Conrad, B., Darioli, R., Schmid, A., & Keller, U. (2015). Health risks associated with meat consumption: a review of epidemiological studies. Int. J. Vitam. Nutr. Res, 85(1-2), 70-78. Wolk, A. (2017). Potential health hazards of eating red meat. Journal of internal medicine, 281(2), 106-122.
Also, the risk reduction of cardiovascular diseases related to aquatic foods is quite well documented to associate with LCPUFA (DHA, EPA) rather than to 'protein' as such, as indicated in line 76-78. It might be that the authors just used 'protein' as a synonym for food, but in this context, it confuses the message, especially because the fatty acids are also covered in the previous objective of reducing deficiencies. The protein reference is further disturbing by inserts mentioning plant proteins (line 77 and line 327-330).	We have removed the reference to plant-based foods on line 76-78, which we agree can be confusing in this particular context and detracts from the main message. However, this particular function of blue foods (reducing cardiovascular disease risk) rests primarily on its capacity to replace and thus reduce the amount of red meat consumed. We recognize that in addition to the potential to reduce cardiovascular disease by replacing red/processed meat, there is some evidence from randomized control trials that omega-3 lowers CVD risk for at-risk groups (e.g. Aung et al 2018; Hu et al, 2019). However, the latest Cochrane review (Abdelhamid et al. 2020) found only moderate and low-certainty evidence for small positive effects of omega-3 intake on cardiovascular health. Given these uncertain effects and the strong evidence of increased risk of CVD mortality associated with red meat consumption (e.g. Abete et al. 2014), we have primarily focused on the role of blue food in replacing red meat rather than its potential health benefits possibly promoted by omega-3. We have however edited the text to also reflect the plausible cardiovascular health benefits of PUFAs.

	Abete, Itziar, Dora Romaguera, Ana Rita Vieira, Adolfo Lopez de Munain, and Teresa Norat. "Association between total, processed, red and white meat consumption and all-cause, CVD and IHD mortality: a meta-analysis of cohort studies." British Journal of Nutrition 112, no. 5 (2014): 762-775. Abdelhamid, Asmaa S., Tracey J. Brown, Julii S. Brainard, Priti Biswas, Gabrielle C. Thorpe, Helen J. Moore, Katherine HO Deane et al. "Omega-3 fatty acids for the primary and secondary prevention of cardiovascular disease." Cochrane Database of Systematic Reviews 11 (2018). Aung, Theingi, Jim Halsey, Daan Kromhout, Hertz C. Gerstein, Roberto Marchioli, Luigi Tavazzi, Johanna M. Geleijnse et al. "Associations of omega-3 fatty acid supplement use with cardiovascular disease risks: meta-analysis of 10 trials involving 77 917 individuals." JAMA cardiology 3, no. 3 (2018): 225-233. Hu, Yang, Frank B. Hu, and JoAnn E. Manson. "Marine omega-3 supplementation and cardiovascular disease: an updated meta-analysis of 13 randomized controlled trials involving 127 477 participants." Journal of the American Heart Association 8, no. 19 (2019): e013543.
The framework builds on analyzing blue food substitution of meat, particularly red meat. If the authors also want to discuss blue food substituting plant-based foods, this should be fully incorporated and include also the nutritional trade-offs (like for omega-3 and B12). Or it should be left out, and not mentioned as it does not add to the context of the analysis.	We believe that we are actually in agreement with the reviewer and that our analysis is not intending to suggest substitution of plant-based proteins with aquatic animal protein. In fact, the times when we mention plant-based protein we did so to make it clear that increasing blue food consumption should be seen as a complementary strategy to already observed food trends of reducing red meat consumption in favor of plant-based protein. In other words, blue foods should be seen as a policy objective to pursue where relevant and justifiable to improve human health, and feasible without large environmentally negative impacts. In the revised version of the paper we have made explicit the fact that our analysis deals with animal-source blue foods. We do not believe removal of all references to plant-based foods is necessary. We feel that in a few select places they serve a purpose to flag that uncritical

expansion and promotion of blue food consumption is not desirable or sustainable. We do this by stating (line 269) that “Substituting all red meats for blue foods is neither feasible nor desirable, and adding or increasing animal-source blue foods to diets of wealthy consumers already rich in animal-source foods would fundamentally undermine the role of blue foods in delivering healthier and less environmentally harmful dietary outcomes”, and in § starting line 406: “However, regardless of how environmentally sustainably produced blue foods are, the global demand for blue foods to address disease and environmental impact in one set of countries (Figure 1 B and C), may reduce the availability and affordability of blue foods for achieving improved nutritional status for vulnerable populations in another (Figure 1 A). Governments can address these tensions by regulating trade and by ensuring that diets incorporating blue foods are considered alongside other means of achieving environmentally sustainable and healthy food system outcomes, such as various forms of more diverse and plant-rich diets”

Additional analysis provided to support our argument relating to sensitivity analysis (see Reviewer 3)
The policy outcome “missing data” was not included in these plots as this category is purely defined by the absence of data and thus not affected by the threshold values.

Supporting Figure 1 Sensitivity results for policy objective “Reducing Deficiencies” along the threshold for SEV B12 (x-axis % of total population). The vertical black line highlights the threshold value used in the analysis. The red point is the number of countries classified into each outcome category in the analysis (at that threshold value). The horizontal line emphasizes the number of countries classified in the analysis. Variation along the x axis provides an indication of the results’ sensitivity to changes in the threshold set for this variable. Variation in the y axis is caused by different thresholds being selected for the other variables relevant to this policy. The white space is caused by a step in the distribution of countries for omega-3 deficiency. When the omega-3 thresholds increase beyond this threshold, a number of countries with equal deficiency values all become relevant together so the number of countries jumps (this can be seen in the step change along the x-axis in the omega three sensitivity (Figure S6 – new histograms)).

Supporting Figure 2 Sensitivity results for policy objective “Reducing Deficiencies” along the threshold for SEV omega-3 (x-axis: % of total population). The vertical black line highlights the threshold value used in the analysis. The red point is the number of countries classified into each outcome category in the analysis (at that threshold value). The horizontal line emphasizes the number of countries classified in the analysis.

Supporting Figure 3 Sensitivity results for policy objective “Reducing Deficiencies” along the threshold for Blue food availability (x-axis kg/cap/yr). The vertical black line highlights the threshold value used in the analysis. The red point is the number of countries classified into each outcome category in the analysis (at that threshold value). The horizontal line emphasizes the number of countries classified in the analysis.

Supporting Figure 4 Sensitivity results for policy objective “Reducing cardiovascular disease risk” along the threshold for red meat consumption (x-axis: g/cap/day). The vertical black line highlights the threshold value used in the analysis. The red point is the number of countries classified into each outcome category in the analysis (at that threshold value). The horizontal line emphasizes the number of countries classified in the analysis.

Supporting Figure 5 Sensitivity results for policy objective “Reducing cardiovascular disease risk” along the threshold for Disability Adjusted Life Years (DALYs). The vertical black line highlights the threshold value used in the analysis. The red point is the number of countries classified into each outcome category in the analysis (at that threshold value). The horizontal line emphasizes the number of countries classified in the analysis.

Supporting Figure 6 Sensitivity results for policy objective “Reducing cardiovascular disease risk” along the threshold for blue food availability (x-axis: kg/cap/yr). The vertical black line highlights the threshold value used in the analysis. The red point is the number of countries classified into each outcome category in the analysis (at that threshold value). The horizontal line emphasizes the number of countries classified in the analysis.

Supporting Figure 7 Sensitivity results for policy objective “Reducing environmental footprints of food consumption and production” along the threshold for ruminant meat consumption (x-axis: g/cap/day). The vertical black line highlights the threshold value used in the analysis. The red point is the number of countries classified into each outcome category in the analysis (at that threshold value). The horizontal line emphasizes the number of countries classified in the analysis.

Supporting Figure 8 Sensitivity results for policy objective “Reducing environmental footprints of food consumption and production” along the threshold for blue food availability (x-axis: kg/cap/yr). The vertical black line highlights the threshold value used in the analysis. The red point is the number of countries classified into each outcome category in the analysis (at that threshold value). The horizontal line emphasizes the number of countries classified in the analysis.

Supporting Figure 9 Sensitivity results for policy objective “Safeguarding food system contributions under climate change” along the threshold employment. The vertical black line highlights the threshold value used in the analysis. The red point is the number of countries classified into each outcome category in the analysis (at that threshold value). The horizontal line emphasizes the number of countries classified in the analysis.

Supporting Figure 10 Sensitivity results for policy objective “Safeguarding food system contributions under climate change” along the threshold export revenue (x-axis: % of GDP). The vertical black line highlights the threshold value used in the analysis. The red point is the number of countries classified into each outcome category in the analysis (at that threshold value). The horizontal line emphasizes the number of countries classified in the analysis.

Supporting Figure 11 Sensitivity results for policy objective “Safeguarding food system contributions under climate change” along the threshold consumption reliance (x-axis: reliance ratio). The vertical black line highlights the threshold value used in the analysis. The red point is the number of countries classified into each outcome category in the analysis (at that threshold value). The horizontal line emphasizes the number of countries classified in the analysis.

Supporting Figure 12 Sensitivity results for policy objective “Safeguarding food system contributions under climate change” along the threshold Climate hazard (x-axis: climate hazard score). The vertical black line highlights the threshold value used in the analysis. The red point is the number of countries classified into each outcome category in the analysis (at that threshold value). The horizontal line emphasizes the number of countries classified in the analysis.

Reviewer Reports on the First Revision:

Referees' comments:

Referee #1 (Remarks to the Author):

The authors have successfully addressed my concerns and the paper could be published.

Referee #2 (Remarks to the Author):

A. There is a great deal of potential for blue foods to reduce environmental impact while also increasing the health of communities globally.

B. This work is novel, interesting, and needed.

C. I am satisfied with the data approach and presentation.

D. These are used appropriately.

E. These are reasonable based on the work presented.

F. My previous comments have been satisfied by the revised version of the manuscript.

G. These are appropriate.

H. This is an interesting manuscript, and I look forward to seeing where the authors take this work.

Referee #3 (Remarks to the Author):

The revised version of manuscript Nature2021-02-03260B looks improved. The authors dealt with all the points raised in my first assessment, and attempted to address thoroughly the key critiques/suggestions. A detailed rebuttal letter is appended, resuming changes made to the text, and addressing the most important issues included in the revision. In particular, the methodology is now more accessible for the reader. This was achieved in two steps: i) by detailing the origins of the methodology used, and associated assumptions in the main text; ii) by providing additional materials in supplementary S1, describing adjustments of the selected methodology. As regards the sensitivity analysis (SA), the authors performed an interesting attempt at approaching the problem at a global level, but finally concluded that pursuing this line of work for SA would have led to somehow less understandable results. Finally, they opted for using the initial one-a-time methodology, and included a new set of revised plots in S1, separating "relevant" and "highly relevant" categories. I appreciated this attempt, and I agree with the choice of keeping SA more easily accessible.

As stated in my first assessment, I feel that the manuscript can be of high interest, and I confirm this indication. I also feel the authors have treated seriously the points raised in my comments within their revision. I therefore find this manuscript suitable for publication, once all the comments raised by the other reviewers will be addressed.

Referee #4 (Remarks to the Author):

I have inserted my response to the authors after each of replies from the authors below:

1. Thank you for this though provoking suggestion. As an author team, this is something we deliberated on extensively before submitting our paper. In fact, we initially embarked on a collection of policies from around the world to provide examples of policy objectives that try to address some of the noted trade-offs or promote synergies. However, we realized that there were (in our opinion) two significant risks with dwelling too much on these examples or providing too many details. These risks relate to the representativity of the policy examples discussed, and to the risk of directing too much attention to certain policy options and inadvertently appearing to recommend these as predominant or desirable options.

More specifically, we believed there is a real risk of individual countries being singled out and erroneously compared to other countries by readers, especially since the focus and scope of the paper did not allow for a full-scale systematic review of national policies, across all the domains we touch on in the paper.

In other words, by providing a more thorough treatment of a non-systematically sampled set of policies, this particular set could risk being misconstrued as either direct recommendations or as a presentation of some predominant options – when in fact some policy options for addressing synergies or tradeoffs may have inadvertently been omitted. We therefore judged it to be more prudent to leave the suggestions at the higher level of policy goals. Sections 5+6 (on synergies and tradeoffs) provide some examples of broad policy options under each subsection, backed with some references. These were intended to provide exactly that – some broad examples of potential relevance across many different cultural or economic context – without providing too much of a specific country context and thus erroneously putting the focus on that nation or context.

Prompted by Reviewer 4's comments we went back over the text and found a certain imbalance, where examples for some trade-offs were lacking. To ensure that every subsection does in fact have examples flagged, we have now added some (demonstrated in lines 260-265, 274-276, 284-290, 304-315, 331-337, 350-356, 368-376). These have been vetted by co-authors to reflect diversity of policy approaches, but will naturally not represent a systematic selection.

With regards to the overarching aim of the paper, we are grateful that the reviewer recognizes our ambition to

not provide all the answers, but rather provide a first step in bringing together the multiple dimensions of blue foods and present a framework that allows nations to get a broad assessment of relevance across the four blue food policy objectives we focus on. It is our hope that in providing this first broad assessment we can trigger policy makers, policy analysts and scholars alike, to delve deeper and make more thorough and granular analysis of the role blue foods could play for addressing multiple goals in specific national or subnational contexts.

We regret that we cannot provide a more refined decision support tool – which we believe is what the reviewer is asking for, but unfortunately a major constraint lies in the lack of publically available data at both national and subnational levels. These kinds of data may be available in specific countries, or to public policy actors in certain regions and could constitute an excellent continuation of our work, by providing the means to conduct finer grained case-based analysis. However, currently the lack of data means a global analysis, such as the one pursued, here would not be feasible.

Response: Thank you for the clarification

2. We agree with the reviewer that we have been too general in our use of the term micronutrient deficiencies in a number of places. We have worked to address this, and specifically in the section identified by the reviewer, we have specified that the nutrients we are particularly

concerned with here are B12 and omega-3. We have added the justification for this. This justification is based on the fact that while blue foods clearly contribute a range of nutritionally important micronutrients, as shown by e.g. Golden et al (2021), their models (based on the most comprehensive dataset we are aware of) show that by 2030, blue foods may contribute a global average of 27.8% of vitamin B12, and 100% of EPA and DHA fatty acids. DHA+EPA content is usually lower in freshwater fish, but not zero (see e.g. Shepon et al 2022). For other nutrients, such as protein, iron, zinc and vitamin A, the total contribution by blue foods is lower, and deficiencies can arguably also be addressed through consumption of a range of other foods. Furthermore, these two (B12 and omega-3) do play an important role in foetal and child growth/development and risk of deficiencies (such as for B12) are particularly elevated for children and elderly. We hope this also helps to early on in the paper, indicate more clearly why our subsequent analysis is restricted to these two micronutrient deficiencies.

Shepon, A., Makov, T., Hamilton, H. A., Müller, D. B., Gephart, J. A., Henriksson, P. J., ... & Golden, C. D. (2022). Sustainable optimization of global aquatic omega-3 supply chain could substantially narrow the nutrient gap. *Resources, Conservation and Recycling*, 181, 106260.

We have also added the justification to clarify our choice in Table 1

Finally, the omega-3 nomenclature has been corrected throughout the paper and extended data. Thank you for noting this mistake.

Response: The authors arguments for limiting the nutrition dimension of the policy objectives to only two nutrients, vitamin B12 and omega-3 are acceptable, though the presentation of this limitation of the nutrition the policy objective can be more clear, to avoid any risk of misinterpretation that this blue food policy tool represent nutrient insufficiency broadly. The policy objective in Table 1 (and in text) can reflect this more clearly. The limitation to these two nutrients is now mentioned in Table 1, but the objective is still generalized to 'Reducing nutrient deficiencies'. It should be more precise for example 'Reducing blue food specific nutrient deficiencies' (or maybe blue food sensitive nutrient deficiencies). Consequently update Table and text, eg the last column in Table 1 to 'Successfully reducing blue food specific nutrient deficiencies'. Update abstract and text to reflect the nutrition objective is specific and not broad nutrition deficiencies. Heading A in Figure 1 ('Reducing nutrition deficiencies') should be updated. Line 147-148: The objective is here updated to be specific on B12 and omega-3.

3. This is a good and insightful point. The reviewer is right that by effectively lumping the deficiencies of vitamin B12 and omega-3 in the Boolean logic used to assess country relevance all we can say is that that 'relevant/highly relevant' countries are countries that have deficiencies above a certain level, for either or both of these micronutrients. In making the decision to pursue this analytical path a number of considerations were made as we deliberated over the pros and cons. We have tried to outline them below, to explain and justify our choice.

It is true that we could have split this analysis into two identical types of analyses for B12 and omega-3, thus ending up with two sub-analyses for this policy objective. As the reviewer notes, this may have given a somewhat different categorization of national relevance, but most likely the main difference would be in the articulation of how to address the deficiency. I.e. deficiencies in omega-3 would mean a country would need to seek supplies of marine blue food, while deficiencies in vitamin B12 is not constrained to marine species.

Since vitamin B12 and PUFAs have a strong overlap in the health contribution they provide –

notably they are recognized as particularly important for physiological and cognitive development – we argue that for the purpose of this paper they can be said to have a very similar ‘blue food function’. As noted by the reviewer above (and elaborated by us above) the purpose of the paper is not to provide a detailed decision support tool for individual countries. Instead we aim to provide a first broad framework to begin to delineate potential blue food policy relevance to countries based on the functions blue foods can play in a particular context. A country which is flagged as potentially relevant for this particular policy in our analysis would need to check if they are low on omega-3 or vitamin B12 or both. This can be easily done in the interactive tool we provide. Based on this, they would then have to consider if their needs could be met by sourcing any aquatic foods, or specifically marine sourced. In this sense the analysis will not provide a detailed policy response to individual countries, but serves as a first indication.

Building a more sophisticated and detailed model for analysis is hampered by the low resolution of much publically available data reported by countries, and by the diversity in contextual nuances that must be considered in national policy development. For example, deficiencies are likely to differ significantly across population segments, perhaps even subnational regions, and be associated with other factors like socio-economic status, etc. Many of these factors (and their subnational variation) would be important to consider when developing actual policy responses, and the model framework we provide here could never be fine-grained enough to serve this purpose.

In light of these different consideration, and pros and cons, we feel strongly that a simpler approach with as much analytical transparency as possible is to be preferred. We hope it is clear why we chose not to treat the two micronutrients separately, as we wanted to reduce the number of layers of analysis the reader would have to penetrate. In other words, reduce the complexity of the analysis and also the message and ease of interpretation. One could argue that a simple but misleading message does not serve anyone, however we do not feel the message is misleading for two reasons. One is that the noted health contribution is similar, and the second is that any country flagged as relevant for a policy can easily check on their values for underlying variables in the interactive tool.

Response: I completely agree ‘a simple but misleading message does not serve anyone’. It remains in my view to be an odd decision to merge B12 and omega-3. It would in my view it would be a stronger message to separate B12 and omega-3. There are similar physiological implications of deficiency as highlighted by the authors, but there are a lot of differences.

To the argument: ‘A country which is flagged as potentially relevant for this particular policy in our analysis would need to check if they are low on omega-3 or vitamin B12 or both’. This supports it would be more straight forward to separate the two.

I will leave it to the editor to decide if the argument justify to keep the simple message by merging the two nutrients (clearly stated throughout the paper, as suggested above), or ask the authors to rerun the analysis to be nutrient specific. To stay simple, then stick to the omega-3 which is clearly blue food relevant.

4. On lines 74-81 we have now specified the micronutrients in focus, as flagged by the reviewer, and justified this focus, with the hope that this clarifies the focus on these particular nutrients, but also the specific nutritional value addressed.

Response: refer to above

5. Under “Healthy alternatives to terrestrial animal protein” we have now changed the wording to note that (as the reviewer rightly points out) that blue foods can be leveraged to reduce dietary risk factors related to several NCDs, including cardiovascular disease, diabetes and various cancers

(Wolk 2017, Richi et al 2015, Miller et al 2022). We have revised and updated the referencing of this section, and clarified that as cardiovascular disease is among the most commonly cited negative health effects of red meat consumption (Wolk 2017, Richi et al 2015) we use it as an example of how countries can assess the relevance of blue food policies depending on their specific conditions.

Miller, V., Micha, R., Choi, E., Karageorgou, D., Webb, P., & Mozaffarian, D. (2022). Evaluation of the Quality of Evidence of the Association of Foods and Nutrients With Cardiovascular Disease and Diabetes: A Systematic Review. *JAMA network open*, 5(2), e2146705-e2146705.

Richi, E. B., Baumer, B., Conrad, B., Darioli, R., Schmid, A., & Keller, U. (2015). Health risks associated with meat consumption: a review of epidemiological studies. *Int. J. Vitam. Nutr. Res.*, 85(1-2), 70-78.

Wolk, A. (2017). Potential health hazards of eating red meat. *Journal of internal medicine*, 281(2), 106-122.

We have removed the reference to plant-based foods on line 76-78, which we agree can be confusing in this particular context and detracts from the main message.

However, this particular function of blue foods (reducing cardiovascular disease risk) rests primarily on its capacity to replace and thus reduce the amount of red meat consumed. We recognize that in addition to the potential to reduce cardiovascular disease by replacing red/processed meat, there is some evidence from randomized control trials that omega-3 lowers CVD risk for at-risk groups (e.g. Aung et al 2018; Hu et al, 2019). However, the latest Cochrane review (Abdelhamid et al. 2020) found only moderate and low-certainty evidence for small positive effects of omega-3 intake on cardiovascular health. Given these uncertain effects and the strong evidence of increased risk of CVD mortality associated with red meat consumption (e.g. Abete et al. 2014), we have primarily focused on the role of blue food in replacing red meat rather than its potential health benefits possibly promoted by omega-3. We have however edited the text to also reflect the plausible cardiovascular health benefits of PUFAs.

Abete, Itziar, Dora Romaguera, Ana Rita Vieira, Adolfo Lopez de Munain, and Teresa Norat. "Association between total, processed, red and white meat consumption and all-cause, CVD and IHD mortality: a meta-analysis of cohort studies." *British Journal of Nutrition* 112, no. 5 (2014): 762-775.

Abdelhamid, Asmaa S., Tracey J. Brown, Julii S. Brainard, Priti Biswas, Gabrielle C. Thorpe, Helen J. Moore, Katherine HO Deane et al. "Omega-3 fatty acids for the primary and secondary prevention of cardiovascular disease." *Cochrane Database of Systematic Reviews* 11 (2018).

Aung, Theingi, Jim Halsey, Daan Kromhout, Hertz C. Gerstein, Roberto Marchioli, Luigi Tavazzi, Johanna M. Geleijnse et al. "Associations of omega-3 fatty acid supplement use with cardiovascular disease risks: meta-analysis of 10 trials involving 77 917 individuals." *JAMA cardiology* 3, no. 3 (2018): 225-233.

Hu, Yang, Frank B. Hu, and JoAnn E. Manson. "Marine omega-3 supplementation and cardiovascular disease: an updated meta-analysis of 13 randomized controlled trials involving 127 477 participants." *Journal of the American Heart Association* 8, no. 19 (2019): e013543.

Response: The argument for red meat replacement refers to a meta-analysis of observational studies (cohorts) and the arguments for health impact of omega-3 intake refers to meta-analysis of RCT studies. The arguments are not balanced. Based on evidence from observational and RCT, the Global Burden of Diseases reach different conclusions on dietary risks of low omega-3 intake and high meat intake.

Afshin et al (2017). Global, regional, and national comparative risk assessment of 84 behavioural,

environmental and occupational, and metabolic risks or clusters of risks, 1990-2016: A systematic analysis for the Global Burden of Disease Study 2016. *The Lancet*, 390(10100), 1345-1422. [https://doi.org/10.1016/S0140-6736\(17\)32366-8](https://doi.org/10.1016/S0140-6736(17)32366-8)

The authors argues for their assumption that the health implications of aquatic food consumption can be reduced to the health gains of reducing red meat consumption. I do not agree. RCT studies do not consequently show positive impact of LCPUFA/omega-3/seafood on cardiovascular risks, but combining evidence from RCT and observational studies (as in the GBD) supports increased LCPUFA/aquatic foods intake have positive health impact in some populations (and no tendency of negative impact).

I will leave it to the editor to decide if the assumption of health implications of aquatic food consumption can be reduced to impacts of red meat substitution is sufficiently argued for publication.

6. We believe that we are actually in agreement with the reviewer and that our analysis is not intending to suggest substitution of plant-based proteins with aquatic animal protein. In fact, the times when we mention plant-based protein we did so to make it clear that increasing blue food consumption should be seen as a complementary strategy to already observed food trends of reducing red meat consumption in favor of plant-based protein. In other words, blue foods should be seen as a policy objective to pursue where relevant and justifiable to improve human health, and feasible without large environmentally negative impacts. In the revised version of the paper we have made explicit the fact that our analysis deals with animal-source blue foods.

We do not believe removal of all references to plant-based foods is necessary. We feel that in a few select places they serve a purpose to flag that uncritical expansion and promotion of blue food consumption is not desirable or sustainable. We do this by stating (line 269) that "Substituting all red meats for blue foods is neither feasible nor desirable, and adding or increasing animal-source blue foods to diets of wealthy consumers already rich in animal-source foods would fundamentally undermine the role of blue foods in delivering healthier and less environmentally harmful dietary outcomes", and in § starting line 406: "However, regardless of how environmentally sustainably produced blue foods are, the global demand for blue foods to address disease and environmental impact in one set of countries (Figure 1 B and C), may reduce the availability and affordability of blue foods for achieving improved nutritional status for vulnerable populations in another (Figure 1 A). Governments can address these tensions by regulating trade and by ensuring that diets incorporating blue foods are considered alongside other means of achieving environmentally sustainable and healthy food system outcomes, such as various forms of more diverse and plant-rich diets"

Response: Hard to disagree, but without data support the arguments for considering more plant-based substitutions are just statements, and dilutes the scientific and data-driven arguments for establishing a stronger foundation for incorporate blue food in food system policies.

Referee #5 (Remarks to the Author):

Note: I was asked to review the QCA part of the manuscript. I don't have expertise in the subject matter, though I could follow the arguments fairly well because the manuscript is very accessible.

The QCA part of the article serves to classify countries with regard to the relevance of the four blue food policies discussed in the main text. My main point that I elaborate on below: The empirical analysis seems to be sound and it is appropriate for achieving the purpose, but it is *not* a QCA study. This is not a point I would hold against the manuscript because one would only have to rewrite the parts about the design and method without reference to QCA.

Based on the supplement, it seems to be argued that the chosen approach is novel. As I understand it, I don't think it is. The approach in the paper uses conditions to build typologies, which is what has been done many times with and without QCA. The last step in figure S1, which is labelled as backward outcome matching (see below on this), is neither novel because it only seems to involve to assign cases to configurations based on their set-membership scores. This is standard in QCA as a case-based method and to typology-building more generally.

The calibration of sets is described in detail and is transparent. I have two comments on this part:

- It is stated on page 18 of the manuscript that authors of relevant expertise gave their input into the calibration process where necessary. It would be good to know who this was for which type of set.

- The justification for the choice of crisp sets is not fully convincing. The degree of transparency is unrelated to the type of set and could be equally high for fuzzy sets. The lack of guidance on what anchors to choose can be taken as a reason for crisp sets because all other set types require the specification of more anchors, which would mean that the level of uncertainty about the anchor choices would be higher. At some point, the authors might hear criticism that crisp sets maximally reduce the information captured by the underlying continuous variables. This is true, but not so much an issue for QCA based on the traditional argument by Charles Ragin that the question is what the relevant degree of information is. In QCA, more information is not necessarily superior to less information, which is different in quantitative research that usually informs criticism of crisp-set research (as per the conventional statistical recommendation not to dichotomize continuous variables).

The biggest issue I see is that the described procedure is not identical with the procedure that has been implemented. I wasn't sure what exactly was done based on the text, so I looked it up in the well written and annotated R file (very positive that it was available, this was helpful for me). What is missing from the actual analysis is the development of a truth table, which summarizes *all* possible combinations of conditions and their negations. As I understand the code in functions.R from lines 264 to 285, types or combinations of conditions are built manually and selectively and not based on a full-scale truth table. The manual development of types is fine and meets the goals of the study, but it is not the same as building a truth table.

More generally, I am highly skeptical that this is a QCA study. There is some disagreement about what 'QCA' is, but I think the development of a truth table is a minimum requirement. (It is indicative that none of the dedicated QCA packages has been used for the empirical analysis.) Again, this is not a problem for the analysis itself because it contains what it is supposed to contain, it is just not a QCA study. Instead, I would describe this as theory-guided typology building (or maybe it is a taxonomy, I am not sure.) The combinations of conditions that make one of the four policies highly relevant, relevant or less relevant are derived from theory and conceptual considerations and are not derived from the data. If this reasoning process is coherent, which I can't judge, then the goal of the analysis is to find out which country belongs to type theoretically defined policy, as summarized in figure 1. This is standard classification or typology-building, but not QCA research.

Specific comments (ignoring the question of whether this is a QCA study or not):

- The supplement speaks of "backward QCA outcome matching". Being familiar with QCA, I don't know what this is supposed to mean, in particular the "backward" part (see also above).
- Personally, I am fine with speaking of independent and dependent variables in a QCA study, but it is common knowledge among QCA users that QCA is different from quantitative research and that one works with sets, conditions and an outcome and not with variables. The use of "variable" tends to trigger QCA methodologists and applied researchers and might not be used here not to draw unnecessary criticism of the terminology.
- The second step of the described procedure does not match the data table with the outcome data. After having calibrated all sets for all cases, one builds a truth table by aggregating over the

calibrated case data. The conditions and outcome data are never separated and there is not matching going on, not even in some informal understanding.

- "asymmetric causality" is misrepresented at one point: It means that the solution for an outcome is most of the time not the negated solution (or inverse solution) for the opposite outcome.

Author Rebuttals to First Revision:

Response to reviewers' comments

Referee #1 (Remarks to the Author):

The authors have successfully addressed my concerns and the paper could be published.

Response: thank you, we are grateful for constructive comments that have vastly improved the manuscript

Referee #2 (Remarks to the Author):

A. There is a great deal of potential for blue foods to reduce environmental impact while also increasing the health of communities globally.

B. This work is novel, interesting, and needed.

C. I am satisfied with the data approach and presentation.

D. These are used appropriately.

E. These are reasonable based on the work presented.

F. My previous comments have been satisfied by the revised version of the manuscript.

G. These are appropriate.

H. This is an interesting manuscript, and I look forward to seeing where the authors take this work.

Response: thank you, we are grateful for constructive comments that have vastly improved the manuscript

Referee #3 (Remarks to the Author):

The revised version of manuscript Nature2021-02-03260B looks improved. The authors dealt with all the points raised in my first assessment, and attempted to address thoroughly the key critiques/suggestions. ... As regards the sensitivity analysis (SA), the authors performed an interesting attempt at approaching the problem at a global level, but finally concluded that pursuing this line of work for SA would have led to somehow less understandable results. Finally, they opted for using the initial one-a-time methodology, and included a new set of revised plots in S1, separating “relevant” and “highly relevant” categories. I appreciated this attempt, and I agree with the choice of keeping SA more easily accessible.

As stated in my first assessment, I feel that the manuscript can be of high interest, and I confirm this indication. I also feel the authors have treated seriously the points raised in my comments within their revision. I therefore find this manuscript suitable for publication, once all the comments raised by the other reviewers will be addressed.

Response: thank you, we are grateful for constructive comments that have vastly improved the manuscript

Referee #4 (Remarks to the Author):

Please note that we have only included the parts of the referee's comments that articulate key opinions,

questions, and requests requiring a response and amendment by us, as to a means to put our response in context.

Referee: The authors arguments for limiting the nutrition dimension of the policy objectives to only two nutrients, vitamin B12 and omega-3 are acceptable, though the presentation of this limitation of the nutrition the policy objective can be more clear, to avoid any risk of misinterpretation that this blue food policy tool represent nutrient insufficiency broadly. The policy objective in Table 1 (and in text) can reflect this more clearly. The limitation to these two nutrients is now mentioned in Table 1, but the objective is still generalized to ‘Reducing nutrient deficiencies’ . It should be more precise for example ‘Reducing blue food specific nutrient deficiencies’ (or maybe blue food sensitive nutrient deficiencies). Consequently update Table and text, eg the last column in Table 1 to ‘Successfully reducing blue food specific nutrient deficiencies’ . Update abstract and text to reflect the nutrition objective is specific and not broad nutrition deficiencies.

Heading A in Figure 1 (‘Reducing nutrition deficiencies’) should be updated.

Line 147-148: The objective is here updated to be specific on B12 and omega-3.

Response: Heeding the call and suggestion of the reviewer we have specified, throughout the paper that we are addressing blue food sensitive nutrient deficiencies. In Table 1, the Blue Food Policy Objective relating to deficiencies has been changed to “**Reducing blue food sensitive nutrient deficiencies**”, and changes have also been made to text pertaining to that row, e.g. when delineating how to leverage blue food functions to address food system challenges.

Figure 1 A has been changed to reflect the new analysis where deficiencies of vitamin B12 and omega-3 are treated separately (see next comment and response). The heading has thus been changed to “Reducing blue food sensitive deficiencies”, with vitamin B12 and omega-3 as subheadings for respective plot.

Figure 2 has been similarly changed to reflect the new analysis where deficiencies of vitamin B12 and omega-3 are treated separately.

Referee: It remains in my view to be an odd decision to merge B12 and omega-3. It would in my view it would be a stronger message to separate B12 and omega-3. There are similar physiological implications of deficiency as highlighted by the authors, but there are a lot of differences. To the argument: ‘A country which is flagged as potentially relevant for this particular policy in our analysis would need to check if they are low on omega-3 or vitamin B12 or both’. This supports it would be more straight forward to separate the two.

I will leave it to the editor to decide if the argument justify to keep the simple message by merging the two nutrients (clearly stated throughout the paper, as suggested above), or ask the authors to rerun the analysis to be nutrient specific. To stay simple, then stick to the omega-3 which is clearly blue food relevant.

Response: We have taken the referee’s comments to heart and have redone the analysis, splitting the variables B12 and omega-3 and treating these as separate. This means Figures 1 and 2 now have been revised. Given the new analysis, we have also revised the text pertaining to the discussion of these

variables in section on “National relevance of blue food policy objectives”, and in the section on “Overlap in policy relevance within countries.” See lines 167-171, and 197-208

Naturally, we have also re-done the sensitivity analysis accordingly, and revised the supplementary material, again splitting out B12 and omega-3.

If our submission is accepted for publication, we will also revise the interactive web-based application to display the split analysis of B12 and omega-3.

Referee (with reference to how the evidence base for various health benefits of blue foods should be discussed): The argument for red meat replacement refers to a meta-analysis of observational studies (cohorts) and the arguments for health impact of omega-3 intake refers to meta-analysis of RCT studies. The arguments are not balanced. Based on evidence from observational and RCT, the Global Burden of Diseases reach different conclusions on dietary risks of low omega-3 intake and high meat intake.

Afshin et al (2017). Global, regional, and national comparative risk assessment of 84 behavioural, environmental and occupational, and metabolic risks or clusters of risks, 1990-2016: A systematic analysis for the Global Burden of Disease Study 2016. *The Lancet*, 390(10100), 1345-1422.

[https://doi.org/10.1016/S0140-6736\(17\)32366-8](https://doi.org/10.1016/S0140-6736(17)32366-8)

Response: We would maintain that the evidence base for health benefits from red meat substitution is currently stronger than the scientific evidence for health benefits related to omega-3 consumption. However, we do agree with the reviewer that it is important to acknowledge the growing evidence of health benefits of DHA and EPA. We also feel that it would not be productive for the paper (and would potentially detract from the key focus) to draw too much attention to the academic debate around the strength of evidence for PUFAs, as in essence they ARE a healthy addition to most diets. We have therefore decided to remove text highlighting current uncertainties (line 89-90 in revised text).

The text now reads: “In this context, the health-promoting role of blue foods rests on the assumption that they can displace some red meat consumption^{1,10} and on the possible health contributions, e.g. of DHA and EPA from aquatic foods^{40,41.}”

We hope this settles this point and thank the reviewer for pushing us to engage in a fruitful discussion around this.

Referee (with reference to the referral to plant-based foods in the manuscript): Hard to disagree, but without data support the arguments for considering more plant-based substitutions are just statements, and dilutes the scientific and data-driven arguments for establishing a stronger foundation for incorporate blue food in food system policies.

Response: Naturally, our study has not specifically investigated the role of plant-based food in food system policies given our focus on blue foods. But we maintain that anyone considering how to grapple with improving the sustainability of our food system – and thus interpreting our analysis and results - needs to consider blue foods in relation to other options, including plant based alternatives. We would maintain that there is ample scientific evidence for the important role of plant based alternatives in

reducing the environmental impacts from food production and consumption in general. For example, in their comprehensive meta-analysis of five environmental indicators, published in Science in 2018, Poore, J., and T. Nemecek conclude that “Most strikingly, impacts of the lowest-impact animal products typically exceed those of vegetable substitutes, providing new evidence for the importance of dietary change”. In addition to this, numerous modeling efforts have been conducted, published in well-respected journals, consistently pointing to the central role of plant based foods for achieving both healthy and environmentally sustainable diets that do not transgress planetary boundaries. A sample of these are referenced in conjunction with our statement. We do not feel that this can be dismissed as unsupported statements, and again – given the important role of considering blue food policy options in relation to other types of food - we argue for keeping the sentence (starting 382) as it is.

Poore, J., and T. Nemecek (2018) Reducing food’s environmental impacts through producers and consumers. Science 360(6392):987–992.

Referee #5 (Remarks to the Author):

Referee: I was asked to review the QCA part of the manuscript. I don’t have expertise in the subject matter, though I could follow the arguments fairly well because the manuscript is very accessible. The QCA part of the article serves to classify countries with regard to the relevance of the four blue food policies discussed in the main text. My main point that I elaborate on below: The empirical analysis seems to be sound and it is appropriate for achieving the purpose, but it is **not** a QCA study. This is not a point I would hold against the manuscript because one would only have to rewrite the parts about the design and method without reference to QCA.

Response: We thank the referee for engaging so constructively with our work. We agree that our methodology is not a QCA, and in an effort to reference a well-established and named methodology, our text mistakenly tied it too closely to QCA. We have revised the methods section and the supplementary material to describe only what we have done, and only make a fleeting reference to QCA as related method from which we derive guidance for best practice for developing crisp sets. Below we elaborate in more detail how we address each of the referee’s comments.

Referee: Based on the supplement, it seems to be argued that the chosen approach is novel. As I understand it, I don’t think it is. The approach in the paper uses conditions to build typologies, which is what has been done many times with and without QCA. The last step in figure S1, which is labelled as backward outcome matching (see below on this), is neither novel because it only seems to involve to assign cases to configurations based on their set-membership scores. This is standard in QCA as a case-based method and to typology-building more generally.

Response: We have revised the language used to describe our approach and, adopting the terminology of the reviewer, we now describe the approach as “theory and expert-guided typology building” (see

first § of methods). We believe the revised text, precludes the need for former Figure S1 and have removed it from the SI.

Referee: The calibration of sets is described in detail and is transparent. I have two comments on this part:

- It is stated on page 18 of the manuscript that authors of relevant expertise gave their input into the calibration process where necessary. It would be good to know who this was for which type of set.

Response: We have attempted to address this request by adding a column in Table S1 where we specify the type of disciplinary expertise leveraged to assess cut-offs for each variable.

Referee: The justification for the choice of crisp sets is not fully convincing. The degree of transparency is unrelated to the type of set and could be equally high for fuzzy sets. The lack of guidance on what anchors to choose can be taken as a reason for crisp sets because all other set types require the specification of more anchors, which would mean that the level of uncertainty about the anchor choices would be higher. At some point, the authors might hear criticism that crisp sets maximally reduce the information captured by the underlying continuous variables. This is true, but not so much an issue for QCA based on the traditional argument by Charles Ragin that the question is what the relevant degree of information is. In QCA, more information is not necessarily superior to less information, which is different in quantitative research that usually informs criticism of crisp-set research (as per the conventional statistical recommendation not to dichotomize continuous variables).

Response: We regret that we did not articulate ourselves very well, and in seeing the referee's comment we note that this is exactly the point we were trying to make. We now phrase it (in the SI) as:

One key reason for choosing crisp set methodology was that we wanted to maximize the ease of interpretation and potential use. Crisp sets arguably retain less information richness than fuzzy sets (where membership of cases is not binary but assigned as degrees of membership to different categories). However, while partial set membership allows for more information from the underlying data to be maintained, it is also likely to result in situations of partial relevance in the outcome variable (Degree of policy relevance). In other words, one could easily end up in a situation where a country is classified as 33% relevant. This would be exceedingly hard for readers to interpret and act on. In other words, crisp sets were chosen in order for countries to receive a clear classification of relevance (highly relevant, relevant, less relevant and missing data) in our analysis.

Another reason for opting for crisp sets is the above noted lack of scientific consensus to guide the exact cut-offs for all variables assessing the conditions. Fuzzy set analysis requires multiple such decisions to be made as each variable is divided into a minimum of three sets (as opposed to a case simply being in the set=1, or out=0), and would have thus increased the uncertainty of the analysis.

Again, our primary goal was ease of interpretation, backed up by transparency in set delineation, conditions and variables used, and we hope the referee feels that this explanation for our choice of crisp sets is more satisfactory. We have revised the text in the methods and supplementary materials to reflect this and provided an expanded justification in Supplementary Information.

Referee: The biggest issue I see is that the described procedure is not identical with the procedure that has been implemented. I wasn't sure what exactly was done based on the text, so I looked it up in the well written and annotated R file (very positive that it was available, this was helpful for me). What is missing from the actual analysis is the development of a truth table, which summarizes *all* possible combinations of conditions and their negations. As I understand the code in functions.R from lines 264 to 285, types or combinations of conditions are built manually and selectively and not based on a full-scale truth table. The manual development of types is fine and meets the goals of the study, but it is not the same as building a truth table.

Response: As noted in our response above, we agree that the close reference to QCA was not helpful, and have revised this throughout. As such, and since we no longer refer to the approach as an adjusted QCA methodology, the issue of the truth table is no longer relevant. We instead describe our methods as a three-step approach, where:

step 1: uses theory and expert assessments to build data table of conditions that logically explain the relevance or non-relevance of each four policy

step 2: develops Boolean logic solution formulas using AND and OR statements

step 3: matches the Boolean solution formulas to the data table (step 1) to assign nations to categories according to 'Degree of policy relevance'

Referee (Specific comments) (ignoring the question of whether this is a QCA study or not):

- The supplement speaks of "backward QCA outcome matching". Being familiar with QCA, I don't know what this is supposed to mean, in particular the "backward" part (see also above).

Response: As we have now revised the description and terminology of the method this has been removed.

- Personally, I am fine with speaking of independent and dependent variables in a QCA study, but it is common knowledge among QCA users that QCA is different from quantitative research and that one works with sets, conditions and an outcome and not with variables. The use of "variable" tends to trigger QCA methodologists and applied researchers and might not be used here not to draw unnecessary criticism of the terminology.

Response: As we have now revised the description and terminology of the method (not linking it to QCA) we judge the potential issue of using 'variables' as a term to be minimal.

- The second step of the described procedure does not match the data table with the outcome data. After having calibrated all sets for all cases, one builds a truth table by aggregating over the calibrated case data. The conditions and outcome data are never separated and there is not matching going on, not even in some informal understanding.

Response: Again, this became an issue when the method was erroneously tied to closely to the notion of QCA. As noted elsewhere, we recognize that we did not conduct a full truth table and that our method relies on developing Boolean logic solution formulas that are then matched to the data table (step 1) to assign nations to categories according to 'Degree of policy relevance'. We hope this addresses the referee's concern.

- "asymmetric causality" is misrepresented at one point: It means that the solution for an outcome is most of the time not the negated solution (or inverse solution) for the opposite outcome.

Response: We have significantly revised the supplementary material and, in removing much of the text relating to QCA, the parts discussing asymmetric causality have also been removed.

Reviewer Reports on the Second Revision:

Referees' comments:

Referee #4 (Remarks to the Author):

The authors have by the revisions responded to my comments, particularly the concerns regarding the nutrition dimension. The policy message is much more transparent by the clear specification of blue food sensitive nutrients, supported by splitting the analysis into B12 and omega-3. I can recommend the manuscript for publication.

Referee #5 (Remarks to the Author):

I appreciate the detailed and constructive response of the authors. The revision is fine with me with regards to the parts that I have focused on in my review.